# Released bacterial ATP shapes local and systemic inflammation during abdominal sepsis

**Daniel Spari[1,2], Annina Schmid[1,2], Daniel Sanchez-Taltavull[1,2], Shaira Murugan[1,2], Keely Keller[1,2], Nadia Ennaciri[1,2], Lilian Salm[1,2], Deborah Stroka[1,2], Guido Beldi[1,2]\***

[1]Department of Visceral Surgery and Medicine, Inselspital, Bern University Hospital, University Hospital of Bern, Bern, Switzerland; [2]Department for BioMedical Research, Visceral Surgery and Medicine, University Hospital of Bern, Bern, Switzerland

**\*For correspondence:**
guido.beldi@insel.ch

**Competing interest:** The authors declare that no competing interests exist.

**Abstract** Sepsis causes millions of deaths per year worldwide and is a current global health priority declared by the WHO. Sepsis-related deaths are a result of dysregulated inflammatory immune responses indicating the need to develop strategies to target inflammation. An important mediator of inflammation is extracellular adenosine triphosphate (ATP) that is released by inflamed host cells and tissues, and also by bacteria in a strain-specific and growth-dependent manner. Here, we investigated the mechanisms by which bacteria release ATP. Using genetic mutant strains of *Escherichia coli* (*E. coli*), we demonstrate that ATP release is dependent on ATP synthase within the inner bacterial membrane. In addition, impaired integrity of the outer bacterial membrane notably contributes to ATP release and is associated with bacterial death. In a mouse model of abdominal sepsis, local effects of bacterial ATP were analyzed using a transformed *E. coli* bearing an arabinose-inducible periplasmic apyrase hydrolyzing ATP to be released. Abrogating bacterial ATP release shows that bacterial ATP suppresses local immune responses, resulting in reduced neutrophil counts and impaired survival. In addition, bacterial ATP has systemic effects via its transport in outer membrane vesicles (OMV). ATP-loaded OMV are quickly distributed throughout the body and upregulated expression of genes activating degranulation in neutrophils, potentially contributing to the exacerbation of sepsis severity. This study reveals mechanisms of bacterial ATP release and its local and systemic roles in sepsis pathogenesis.

## eLife assessment

This **fundamental** study advances our understanding of the role of bacterial-derived extracellular ATP in the pathogenesis of sepsis. The evidence supporting the conclusions is **compelling**, although not all concerns from a previous round of reviews were adequately addressed. The work will be of broad interest to researchers on microbiology and infectious diseases.

## Introduction

Worldwide, 11 million sepsis-related deaths were reported in 2017, which accounted for an estimated 19.7% of all global deaths (*Rudd et al., 2020*). Given its high incidence and immense socio-economic burden, the World Health Organization (WHO) has declared sepsis as a global health priority (*Reinhart et al., 2017*).

Sepsis is defined as life-threatening organ dysfunction caused by an imbalanced host immune response to infection. Antibiotic treatment remains the main approach to treat sepsis; however, despite the use of broad-spectrum antibiotics, lethality remains high. Immune-modulating strategies are an additional

**eLife digest** Sepsis is a severe condition often caused by the body's immune system overreacting to bacterial infections. This can lead to excessive inflammation which damages organs and requires urgent medical care. With sepsis claiming millions of lives each year, new and improved ways to treat this condition are urgently needed. One potential strategy for treating sepsis is to target the underlying mechanisms controlling inflammation.

Inflamed and dying cells release molecules called ATP (the energy carrier of all living cells), which strongly influence the immune system, including during sepsis. In the early stages of an infection, ATP acts as a danger signal warning the body that something is wrong. However, over time, it can worsen infections by disturbing the immune response.

Similar to human cells, bacteria release their own ATP, which can have different impacts depending on the type of bacteria and where they are located in the body. However, it is not well understood how bacterial ATP influences severe infections like sepsis.

To investigate this question, Spari et al analysed how ATP is released from *Escherichia coli*, a type of bacteria that causes severe infections. This revealed that the bacteria secrete ATP directly in to their environment and via small membrane-bound structures called vesicles.

Spari et al. then probed a mouse model of abdominal sepsis which had been infected with *E. coli* that release either normal or low levels of ATP. They found that the ATP released from *E. coli* impaired the mice's survival and lowered the number of neutrophils (immune cells which are important for defending against bacteria) at the site of the infection. The ATP secreted via vesicles also altered the role of neutrophils but in more distant regions, and it is possible that these changes may be contributing to the severity of sepsis.

These findings provide a better understanding of how ATP released from bacteria impacts the immune system during sepsis. While further investigation is needed, these findings may offer new therapeutic targets for treating sepsis.

approach to antibiotics to curb excessive inflammation and to support an effective defense against the infectious agents. Until recently however, most trials inhibiting cytokine responses or Toll-like receptors failed, indicating that alternative approaches for immunomodulation are required (*Cao et al., 2023*).

It has been shown that adenosine triphosphate (ATP), as soon as it is released into the extracellular space, critically modulates inflammatory and immune responses (*Di Virgilio et al., 2020*; *Eltzschig et al., 2012*) by activating ionotropic P2X and metabotropic P2Y receptors (*Burnstock, 2020*; *Junger, 2011*). Also, such purinergic signaling critically alters immune responses during sepsis (*Dosch et al., 2019*; *Ledderose et al., 2016*; *Spari and Beldi, 2020*). In particular, we have described that the connexin-dependent release of ATP by macrophages initiates an autocrine loop of over-activation, resulting in altered local and systemic cytokine secretion, which exacerbates abdominal sepsis (*Dosch et al., 2019*). Such sepsis-promoting effects of host-derived extracellular ATP are secondary to inflammation initiated by bacterial pathogens.

Recently, it has been discovered that also bacteria release ATP into the extracellular space (*Mempin et al., 2013*). Such ATP release might be a conserved mechanism of protection from host defense and precede host responses. ATP release has been shown for a variety of bacteria including the sepsis-associated *Escherichia coli* (*E. coli*) and *Klebsiella pneumoniae* (*K. pneumoniae*) from the Proteobacteria phylum or *Enterococcus faecalis* (*E. faecalis*) and *Staphylococcus aureus* (*S. aureus*) from the Firmicutes phylum (*Diekema et al., 2019*; *Hironaka et al., 2013*; *Iwase et al., 2010*; *Mempin et al., 2013*; *Mureșan et al., 2018*).

The mechanisms by which inflammatory and immune responses in the host are modulated by such released bacterial ATP have just begun to be elucidated (*Spari and Beldi, 2020*). In colonized compartments such as the intestine, it has been shown that ATP released by mutualistic bacteria modulates local cellular and secretory immune responses (*Atarashi et al., 2008*; *Perruzza et al., 2017*; *Proietti et al., 2019*) and in the mouth, bacterial ATP release leads to biofilm dispersal and periodontitis (*Ding et al., 2016a*; *Ding and Tan, 2016b*). However, the role of ATP released by bacteria invading non-colonized compartments, such as the abdominal cavity or the blood in the context of local and systemic infections, remains to be determined.

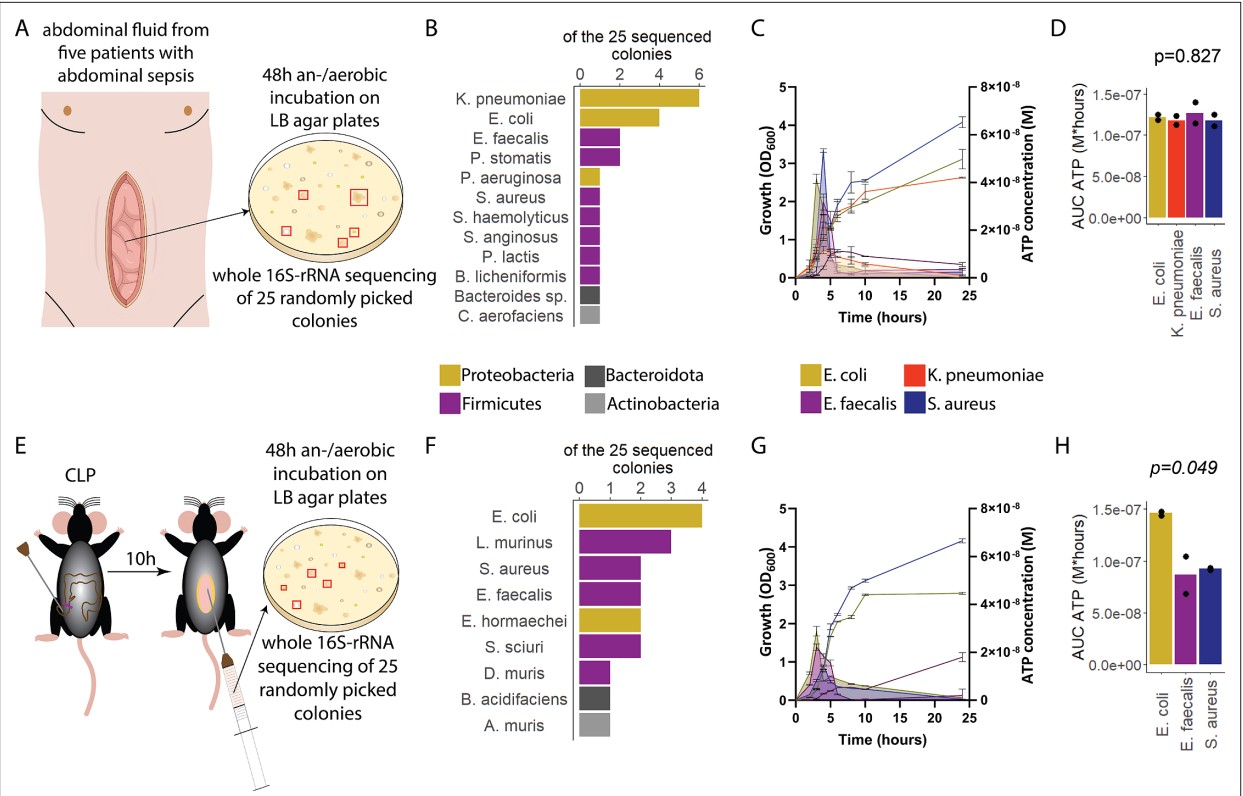

**Figure 1.** Sepsis-associated bacteria release adenosine triphosphate (ATP) in a growth-dependent manner. (**A**) Experimental approach to isolate and cultivate sepsis-associated bacteria from abdominal fluid of patients with abdominal sepsis. (**B**) Bacterial species identified by whole 16S-rRNA Sanger sequencing from abdominal fluid of patients with abdominal sepsis. Three colonies out of 25 could not be identified. (**C**) Measurement of released ATP (M) and growth (OD$_{600}$) over time (hours) from the four sepsis-associated bacteria *E. coli*, *K. pneumoniae*, *E. faecalis,* and *S. aureus* isolated from patients. N=2 independent bacteria cultures. Means and standard deviations are shown. (**D**) Area under the curve (AUC) of released ATP over time (M*hours) of the previously assessed bacteria (cumulative ATP). One-way ANOVA, N=2 independent bacteria cultures. Means and individual values are shown. (**E**) Experimental approach to isolate and cultivate sepsis-associated bacteria from abdominal fluid of mice with abdominal sepsis. (**F**) Bacterial species identified by whole 16S-rRNA Sanger sequencing from abdominal fluid of mice with abdominal sepsis. Seven colonies out of 25 could not be identified. (**G**) Measurement of released ATP (M) and growth (OD$_{600}$) over time (hours) from the three sepsis-associated bacteria *E. coli*, *E. faecalis,* and *S. aureus* isolated from mice. N=2 independent bacteria cultures. Means and standard deviations are shown. (**H**) AUC of released ATP over time (M*hours) of the previously assessed bacteria (cumulative ATP). One-way ANOVA, N=2 independent bacteria cultures. Means and individual values are shown.

The online version of this article includes the following figure supplement(s) for figure 1:

**Figure supplement 1.** Experimental approach to measure released bacterial adenosine triphosphate (ATP) and growth over time.

In this study, we investigated if ATP released from bacteria influences the outcome of abdominal sepsis. We first isolated sepsis-associated bacteria and measured the amount of ATP they release. Second, we analyzed the function of the inner bacterial membrane on ATP release over time in *E. coli* and in *E. coli* with mutations in integral respiratory chain proteins (***Mempin et al., 2013***). Third, the function of the outer bacterial membrane on ATP release during growth was assessed using porin mutants (***Alvarez et al., 2017***; ***Choi and Lee, 2019***). Fourth, we investigated local effects of ATP in the abdominal cavity. Lastly, based on the finding that bacteria secrete outer membrane vesicles (OMV) (***Schwechheimer and Kuehn, 2015***), we investigated systemic consequences of released bacterial ATP.

## Results

### *E. coli*, one of the major pathogens in sepsis, releases ATP in a growth-dependent manner

To assess ATP release of sepsis-associated bacteria, abdominal fluid of patients with abdominal sepsis was sampled and an/aerobically incubated on LB agar plates (*Figure 1A*). Twenty-five different colonies were randomly picked and analyzed by whole 16S-rRNA Sanger sequencing, which resulted in 12 different bacterial species (*Figure 1B*). From these, the four most clinically important sepsis-associated bacteria *E. coli*, *K. pneumoniae* (both gram$^{neg}$), *E. faecalis*, and *S. aureus* (both gram$^{pos}$) (*Diekema et al., 2019*; *Mureșan et al., 2018*) were further cultivated for experimental studies. We quantified released ATP over time (*Figure 1—figure supplement 1*) using a luciferin-luciferase-based assay. A growth-dependent release of ATP was observed in all species, peaking during exponential growth phase (*Figure 1C*). Cumulative amount of released ATP was quantified using the area under the curve (AUC) of released ATP over time (AUC ATP). ATP release was detected across all assessed species (*Figure 1D*). To model these findings in mice, abdominal sepsis was induced using a standardized cecal ligation and puncture (CLP) model (*Dosch et al., 2019*; *Figure 1E*). Similar to human samples, Proteobacteria and Firmicutes phyla were predominating (*Figure 1F*, see *Figure 1B*). In the mouse model, *E. coli* released notably more cumulative ATP compared to *E. faecalis* and *S. aureus* (*Figure 1G and H*). The bacterial species assessed in humans and mice (*E. coli*, *E. faecalis*, *S. aureus*) differed on the strain level (sequences deposited, see Data availability statement) confirming that ATP release is strain-specific (*Mempin et al., 2013*). In summary, sepsis-associated bacteria release ATP in a growth-dependent and strain-specific manner. *E. coli,* which is one of the most frequent facultative pathogens in patients with abdominal sepsis, released the highest amount of ATP isolated from a standardized mouse model of abdominal sepsis.

### Bacterial ATP release is dependent on ATP synthesis at the inner bacterial membrane and correlates with bacterial growth

After having demonstrated that sepsis-associated bacteria release ATP, we next questioned whether and how ATP release is dependent on ATP generation. ATP synthase and cytochrome oxidases, which are located in the inner bacterial membrane, are key components of ATP generation under aerobic conditions. Therefore, ATP release of the *E. coli* parental strain (PS) was compared to all available mutants of ATP synthase subunits (ΔatpA, ΔatpB, ΔatpC, ΔatpD, ΔatpE, ΔatpF, ΔatpH) and cytochrome *bo₃* oxidase subunits (ΔcyoA, ΔcyoB, ΔcyoC, ΔcyoD) from the Keio collection (*Baba et al., 2006*; *Yamamoto et al., 2009*). The use of different mutants allows to identify the most relevant subunits influencing ATP release. Also, it allows to identify a possible correlation of bacterial ATP release with growth, which is a function of ATP generation (*Figure 2A*; *Galber et al., 2021*; *Ihssen et al., 2021*; *Ivancic et al., 2008*; *Lundin, 2000*).

Bacterial growth (OD$_{600}$) and ATP release were measured over time and cumulative ATP release (AUC ATP) was assessed (*Figure 2B*). We first noticed that mutations in subunits of ATP synthase, which is the key enzyme of ATP generation, were generally associated with significantly lower cumulative ATP release compared to mutations in cytochrome *bo₃* oxidase subunits (*Figure 2C*). Therefore, we suspected an interrelation between ATP generation and released ATP (see growth and ATP release curves in *Figure 2B*). We determined cumulative growth (AUC growth) in addition to cumulative ATP release and indeed, cumulative ATP release and cumulative growth were positively correlated (*Figure 2D*), similar to peak ATP and peak growth (OD$_{600}$) that are positively correlated (*Figure 2—figure supplement 1*). In summary, ATP release is directly dependent on ATP generation at the inner bacterial membrane. Mutations in subunits of bacterial ATP synthase have a higher impact on ATP generation, growth, and ATP release than mutations in subunits of cytochrome *bo₃* oxidase (*Figure 2C*).

### Outer bacterial membrane integrity and bacterial death determine bacterial ATP release during growth

We next focused on the outer membrane by challenging its integrity while leaving ATP generation and the inner membrane intact. For that purpose, we used the *E. coli* porin mutants ΔompC, ΔompF, ΔlamB, and ΔphoE (*Figure 3A*; *Baba et al., 2006*; *Yamamoto et al., 2009*), which have been shown

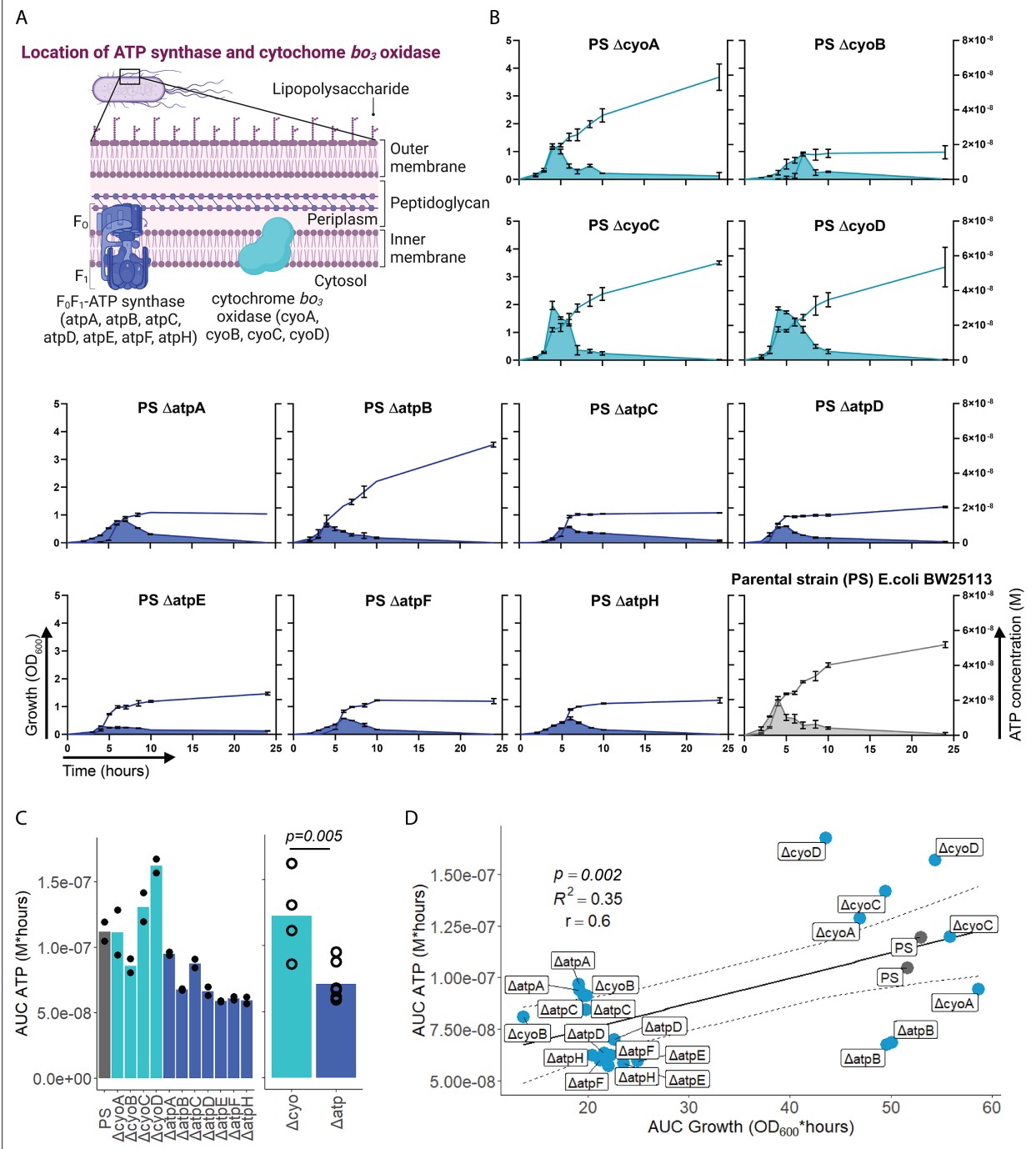

**Figure 2.** Adenosine triphosphate (ATP) release is dependent on ATP synthesis. (**A**) Illustration depicting the location of ATP synthase and cytochrome *bo₃* oxidase in gram[neg] bacteria. (**B**) Measurement of released ATP (M) and growth (OD$_{600}$) over time (hours) from cytochrome *bo₃* oxidase (cyo) and ATP synthase (atp) mutants. The parental strain (PS) was added as a control. N=2 independent bacteria cultures. Means and standard deviations are shown. (**C**) Area under the curve (AUC) of released ATP over time (M*hours) of the previously assessed bacteria (cumulative ATP) is shown individually in the left panel. N=2 independent bacteria cultures. Means and individual values are shown. Means of grouped cyo and atp mutants are compared in the right panel. t-Test. Means and individual values are shown. (**D**) Cumulative ATP (M*hours) and cumulative growth (OD$_{600}$*hours) of all assessed cyo and atp mutants and the PS were plotted against each other. Pearson's correlation (r) and coefficient of determination (R$^2$) of the applied linear model are depicted. 95% confidence level is shown by the black dashed lines.

The online version of this article includes the following figure supplement(s) for figure 2:

*Figure 2 continued on next page*

*Figure 2 continued*

**Figure supplement 1.** Peak ATP (M) and peak growth (OD$_{600}$*hours) of all assessed cyo and atp mutants and the PS were plotted against each other.

to suffer from impaired outer membrane integrity in varying degrees (*Choi and Lee, 2019*). ATP release and growth were measured over time including *E. coli* PS as baseline and membrane destabilizing EDTA and stabilizing Ca$^{2+}$ as additional controls (*Leive, 1968*). Cumulative ATP release (AUC ATP) from the porin mutants were notably different compared to the PS, being lowest in *ΔompC* and highest in *ΔompF* (*Figure 3B and C*).

Interestingly, the *ΔompF* mutant had also a very high peak of released ATP (*Figure 3B*). We hypothesized that this is because of impaired membrane integrity, resulting in ATP release during growth and potentially bacterial death. Thus, we focused on the individual ATP peaks during growth, which were observed after 4 hr of culturing. There was a strong negative correlation between the individual peak of released ATP (marked by the red line in *Figure 3B*) and growth at the same time point (*Figure 3D*).

We did not interfere with ATP generation at the inner membrane but deliberately challenged the outer membrane and tested therewith if destabilization of the outer membrane integrity is associated with bacterial death. Indeed, outer membrane integrity and bacterial death are significantly increased in *ΔompF* compared to *ΔompC* and the PS after 4 hr (ATP peak) of culturing (*Figure 3E and F*), akin to the amount of released ATP (*Figure 3G*). We conclude from these data that destabilization of the outer bacterial membrane (as observed with the *ΔompF* mutant) results in bacterial death that is associated with ATP release.

In summary, outer membrane integrity and finally bacterial death notably contribute to the amount of bacterial ATP release during growth.

## Released bacterial ATP reduces neutrophil counts and impairs survival during abdominal sepsis

Next, we wanted to investigate the function of bacterial ATP release in vivo. To study this, we transformed the *E. coli* PS with an arabinose-inducible apyrase (PS+pAPY) and compared it to the PS transformed with the empty vector (PS+pEMPTY) (*Proietti et al., 2019*). In this model, ATP released by bacteria is hydrolyzed and consequently depleted by a periplasmic apyrase (*Santapaola et al., 2006*; *Scribano et al., 2014*).

Indeed, apyrase induction resulted in a significant reduction of ATP release in PS+pAPY, compared to PS+pEMPTY (*Figure 4—figure supplement 1A and B*). To test the consequences in vivo, apyrase was induced by arabinose 3 hr before intraabdominal (i.a.) injection into wild type C57Bl/6 mice (*Figure 4A*). Thereby ATP release was abrogated in the bacteria cultures that were used for injection (*Figure 4B*). In vivo, no difference in ATP levels was detected when ATP was measured directly in the abdominal fluid after 4 (*Figure 4C*) and 8 hr (*Figure 4—figure supplement 1C*). This is not surprising given that ATP is rapidly hydrolyzed by ectonucleotidases in vivo (*Eltzschig et al., 2012*). After both 4 (*Figure 4D and E*) and 8 hr (*Figure 4—figure supplement 1D and E*), no differences in local or systemic bacterial counts were observed. Yet, despite similar bacterial counts, the survival was significantly higher in the absence of bacterial ATP (PS+pAPY) compared to ATP-generating controls (PS+pEMPTY) after i.a. injection (*Figure 4F*).

Next, we asked how this difference in bacterial ATP release affects the immune system. Therefore, inflammatory cells in the abdominal cavity were characterized using flow cytometry (*Figure 4G*). As expected, a disappearance reaction of large peritoneal macrophages (LPM) was observed after both, *E. coli* PS+pEMPTY and *E. coli* PS+pAPY, i.a. injection compared to sham controls after 4 (*Figure 4H*) and 8 hr (*Figure 4—figure supplement 1F*; *Ghosn et al., 2010*). Such LPM disappearance following abdominal *E. coli* infection is a result of free-floating clots composed of LPM and neutrophils and important for effective pathogen clearance (*Salm et al., 2023*; *Vega-Pérez et al., 2021*; *Zindel et al., 2021*) but not dependent on ATP release according to our data. Interestingly, however, the number of small peritoneal macrophages (SPM) and CX3CR1[pos] monocytes was significantly reduced, whereas neutrophils were significantly increased almost up to 8 hr (*Figure 4I*, *Figure 4—figure supplement 1G*) in bacterial ATP-depleted abdominal sepsis (PS+pAPY). This effect is dependent on released bacterial ATP given that no differences in bacterial counts were observed (see *Figure 4D and E*, *Figure 4—figure supplement 1D–E*).

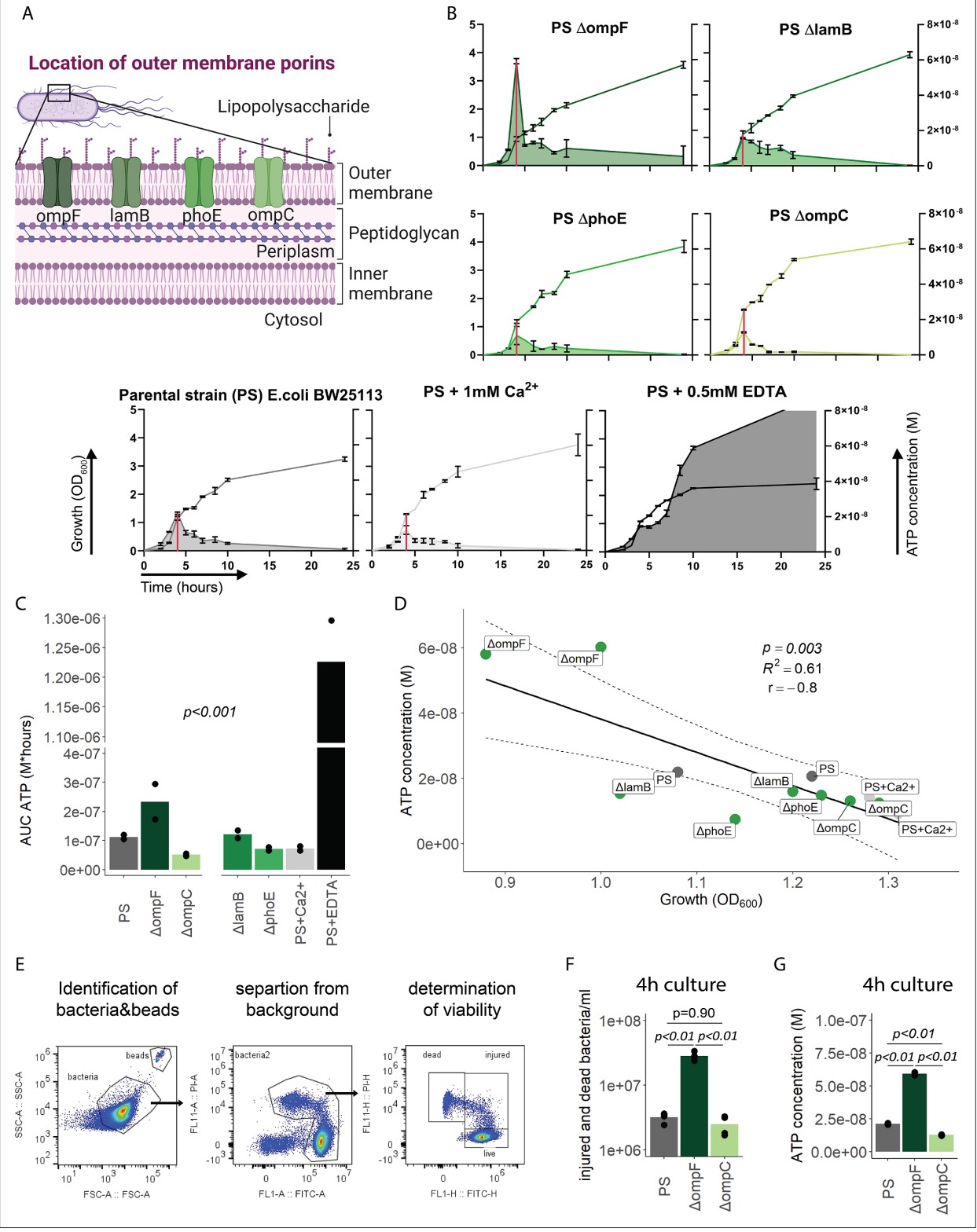

**Figure 3.** Outer membrane integrity and bacterial death determine bacterial adenosine triphosphate (ATP) release during growth. (**A**) Illustration depicting the location of outer membrane porins in gram[neg] bacteria. (**B**) Measurement of released ATP (M) and growth ($OD_{600}$) over time (hours) from outer membrane porin mutants. The parental strain (PS) and the PS supplemented with either 1 mM $Ca^{2+}$ or 0.5 mM EDTA were added as controls. N=2 independent bacteria cultures. Means and standard deviations are shown. The red line marks the individual peak of ATP release and growth ($OD_{600}$) at that time point. (**C**) Area under the curve (AUC) of released ATP over time (M*hours) of the previously assessed bacteria (cumulative ATP). One-way

*Figure 3 continued on next page*

*Figure 3 continued*

ANOVA, N=2 independent bacteria cultures. Means and individual values are shown. (**D**) ATP concentration (M) and growth (OD$_{600}$) at the individual peak of ATP release of all assessed outer membrane porin mutants, the PS, and the PS+Ca$^{2+}$ (no peak for the EDTA control) were plotted against each other. Pearson's correlation (**r**) and coefficient of determination (R$^2$) of the applied linear model are depicted. 95% confidence level is shown by the black dashed lines. (**E**) Gating strategy to identify added counting beads, live, injured, and dead bacteria. (**F**) Quantitative assessment of injured and dead bacteria, as identified by flow cytometry after 4 hr of culturing (ATP peak) of the PS, ΔompF and ΔompC. One-way ANOVA followed by Tukey post hoc test, N=4 independent bacteria cultures. Means and individual values are shown. (**G**) ATP concentration (M) after 4 hr of culturing (ATP peak) of the PS, ΔompF and ΔompC. One-way ANOVA followed by Tukey post hoc test, N=2 independent bacteria cultures. Means and individual values are shown.

In summary, ATP released by bacteria suppresses abdominal inflammatory responses and worsened survival in a model of abdominal sepsis.

## Establishing ATP-loaded OMV as a model system to assess the systemic relevance of bacterial ATP

ATP is rapidly metabolized in the extracellular space and therefore, the mode of action of released bacterial ATP is limited to the immediate cellular vicinity (*Junger, 2011*). However, the outcome of sepsis is not only dependent on local but also on systemic responses to microorganisms. Therefore, we hypothesized that bacterial ATP has systemic effects as protected cargo in OMV (*Alvarez et al., 2017*). OMV are small (20–300 nm) spherical particles that are released by both gram$^{neg}$ and gram$^{pos}$ bacteria (*Schwechheimer and Kuehn, 2015*). In gram$^{neg}$ bacteria, they bulge off the outer membrane and disseminate throughout the body (*Jang et al., 2015*). They are equipped with typical bacterial surface features lacking the machinery for self-reproduction, and contain DNA, proteins, and metabolites (*Baeza and Mercade, 2021*; *Bitto et al., 2017*; *Kulp and Kuehn, 2010*; *Lee et al., 2007*). Therefore, OMV are suited as a systemic delivery system for bacterial ATP. Indeed, recently, ATP has been detected in OMV derived from pathogenic *Neisseria gonorrhoeae*, *Pseudomonas aeruginosa* PAO1, and *Acinetobacter baumannii* AB41 (*Pérez-Cruz et al., 2015*).

To assess the potential of OMV as ATP carriers, we compared the OMV production from several hypervesiculation *E. coli* mutants (ΔmlaE, ΔmlaA, ΔrfaD, ΔdegP, ΔrodZ, ΔnlpI, ΔtolB) (*Figure 5A*; *McBroom et al., 2006*). The ΔnlpI and ΔtolB strains showed a 20- and 30-fold increase of OMV when compared with the PS (*Figure 5B*). We then assessed ATP release and growth over time from ΔnlpI, ΔtolB, and the PS, to identify their individual peak of ATP release, and isolated OMV at their individual peak of ATP release and after 24 hr (*Figure 5—figure supplement 1A*). ATP was detected in OMV from all assessed strains at the individual peak of ATP release but only in minimal detectable levels after 24 hr (*Figure 5C*). The ΔtolB OMV isolated after 24 hr were then used as ATP-depleted vehicles. Density gradient ultracentrifugation showed that most ΔtolB OMV were of similar density and protein composition (*Figure 5D*, *Figure 5—figure supplement 1B*). They were equipped with outer membrane ompF but not cytoplasmic ftsZ (*Figure 5D*), indicating that they are outer membrane derived. In order to generate OMV with known and constant ATP concentrations, we used electroporation (EP) to load OMV with ATP while empty OMV (ΔtolB OMV harvested from 24 hr culture) served as ATP-depleted controls (*Fu et al., 2020*; *Lennaárd et al., 2021*). Before and after EP, the OMV size distribution was assessed by nanoparticle tracking analysis and morphology by electron microscopy (*Figure 5E*, *Figure 5—figure supplement 1C*). OMV were loaded with ATP (*Figure 5F*) and over time, the amount of ATP in OMV decreased, especially at physiological (37°C) temperature (*Figure 5G*) as opposed to 4°C (*Figure 5—figure supplement 1D*).

In summary, OMV contain ATP and release ATP at physiological temperatures. To use OMV as an ATP delivery system, empty OMV were loaded using EP.

## OMV-derived bacterial ATP induces degranulation processes in neutrophils after lysosomal uptake

OMV are potent inducers of inflammation and sepsis (*Park et al., 2010*; *Park et al., 2013*), which travel throughout the body and are taken up by a variety of cells (*Bittel et al., 2021*; *Kim et al., 2013*; *Lee et al., 2018*). To tests the hypothesis that ATP within OMV mediates systemic effects of invasive

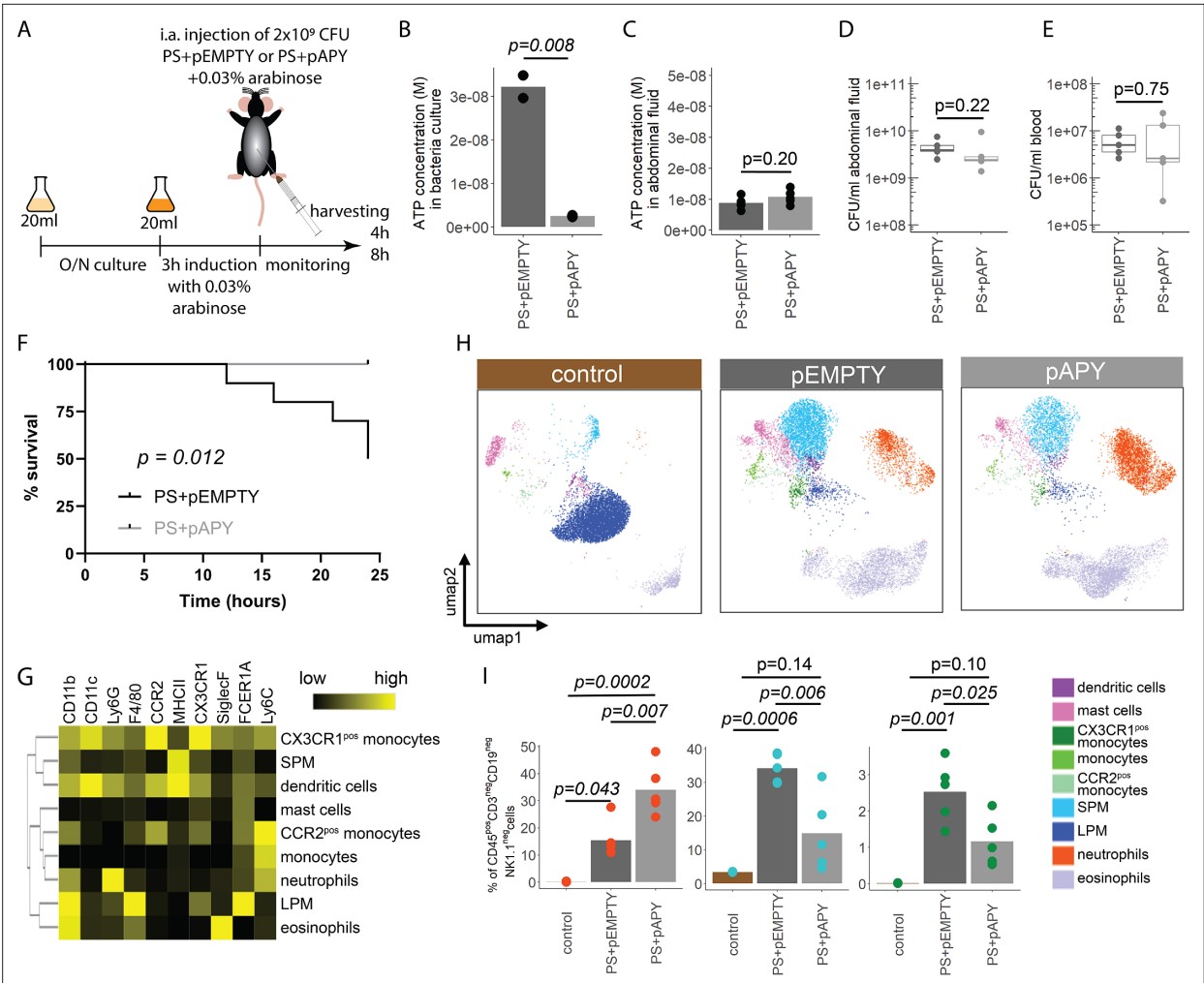

**Figure 4.** Bacterial adenosine triphosphate (ATP) reduces neutrophil counts and impairs sepsis outcome in vivo. (**A**) Experimental approach to determine the local role of bacterial ATP in vivo, intraabdominal (i.a.) injecting parental strain (PS)+pEMPTY or PS+pAPY. (**B**) Measurement of released ATP (M) in bacteria culture supernatant immediately before bacteria were i.a. injected. t-Test, N=2 independent bacteria cultures. Means and individual values are shown. (**C**) Measurement of ATP (M) in abdominal fluid from mice 4 hr after i.a. injection of bacteria. t-Test, n=5 animals per group of N=2 independent experiments. Means and individual values are shown. (**D**) Quantitative assessment of colony forming units in abdominal fluid and (**E**) blood from mice 4 hr after i.a. injection of bacteria. Wilcoxon rank-sum test, n=5 animals per group of N=2 independent experiments. Means and individual values are shown. No growth for controls was detected. (**F**) Kaplan-Meier curves of mice after i.a. injection of bacteria. Log-rank test, n=10 animals per group. (**G**) Heatmap showing surface marker expression (x-axis), which was used to characterize the different immune cell populations (y-axis). (**H**) Concatenated (n=5 animals for each treatment group, n=3 animals for control group of N=2 independent experiments) and down-sampled images of immune cell populations characterized in the abdominal cavity 4 hr after sham treatment or i.a. injection of bacteria. (**I**) Abundance of neutrophils, small peritoneal macrophages (SPM), and CX3CR1$^{pos}$ monocytes in abdominal fluid from mice 4 hr after sham treatment or i.a. injection of bacteria. One-way ANOVA followed by Tukey post hoc test, n=5 animals for each treatment group, n=3 animals for control group of N=2 independent experiments. Means and individual values are shown.

The online version of this article includes the following figure supplement(s) for figure 4:

**Figure supplement 1.** Immune cell characterization 8 hr after intraabdominal (i.a.) injection of bacteria.

bacteria, we injected ATP-loaded and empty OMV i.a. and investigated the resulting inflammation 1 hr after injection (**Figure 6A**).

In the abdominal fluid, uptake of DiI-stained OMV by leukocytes was independent of ATP cargo (**Figure 6B**, **Figure 6—figure supplement 1**) and all major cell populations with phagocytic ability (LPM, SPM, neutrophils) were highly positive for OMV (**Figure 6—figure supplement 2A and B**). LPM dramatically decreased, whereas neutrophils increased in response to OMV when compared to sham controls (**Figure 6—figure supplement 2C**).

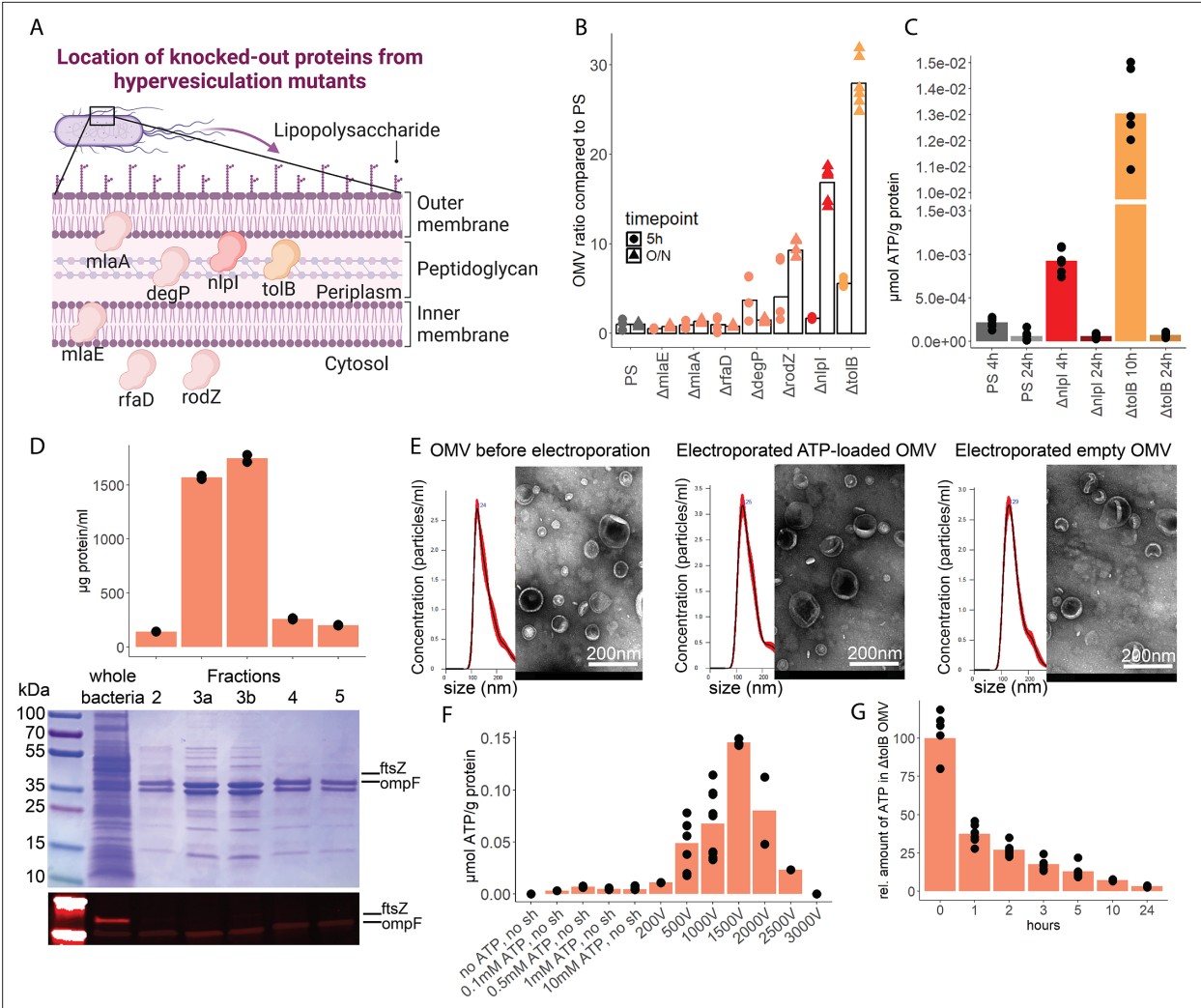

**Figure 5.** Outer membrane vesicles (OMV) contain adenosine triphosphate (ATP) and can be exploited as a model to assess the systemic relevance of bacterial ATP. (**A**) Illustration depicting the location of assessed proteins that lead to a hypervesiculation phenotype if knocked out in the gram[neg] bacterium *E. coli*. (**B**) Relative amount of OMV compared to the parental strain (PS) isolated from growth cultures of the assessed hypervesiculation mutants after 5 hr (exponential growth phase) and O/N (stationary phase). n=2 measurements of N=3 independent bacteria cultures. Means and individual values are shown. (**C**) Absolute quantification of ATP in OMV isolated from growth cultures of the PS, *ΔnlpI* and *ΔtolB* at their individual peak of ATP release and after 24 hr. n=2 measurements of N=3 independent bacteria cultures. Means and individual values are shown. (**D**) Amount of protein (BCA assay) detected in different fractions after density gradient ultracentrifugation. n=2 measurements of the different fractions. 20 µl of *E. coli* growth culture and 20 µl of each fraction were then characterized by Coomassie blue staining and specific detection of outer membrane ompF and cytoplasmic ftsZ. (**E**) Characterization of OMV by nanoparticle tracking analysis (n=5 measurements per sample) and electron microscopy (representative image) before and after electroporation. (**F**) Absolute quantification of ATP in OMV, which were loaded using different strategies. Columns 2–5: different concentrations of ATP incubated for 1 hr at 37°C (passive filling). Columns 6–12: different voltages with fixed settings for resistance (100 Ω) and capacitance (50 µF). N=2–9 independent experiments. Means and standard deviations are shown. (**G**) Relative quantification of ATP in OMV over 24 hr at 37°C after electroporation (0 hr=100%). n=2 measurements of N=3 independent experiments. Means and individual values are shown.

The online version of this article includes the following figure supplement(s) for figure 5:

**Figure supplement 1.** Adenosine triphosphate (ATP) measurement of the parental strain (PS), *ΔnlpI* as well as *ΔtolB* and outer membrane vesicle (OMV) collection and characterization.

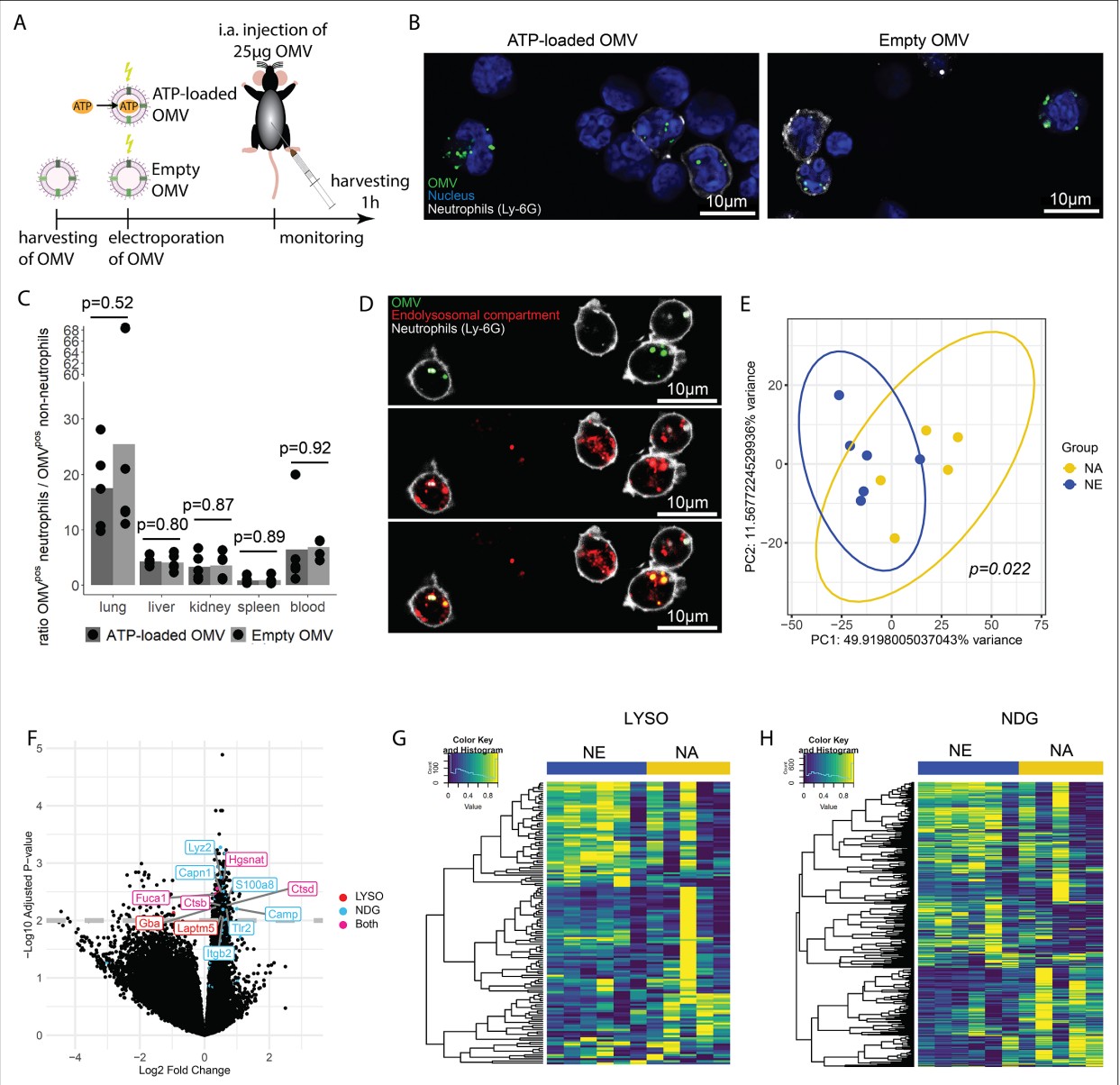

**Figure 6.** Bacterial adenosine triphosphate (ATP) within outer membrane vesicles (OMV) upregulates lysosome-related pathways and degranulation processes in neutrophils. (**A**) Experimental approach to determine the systemic role of bacterial ATP in vivo, intraabdominal (i.a.) injecting ATP-loaded or empty OMV. (**B**) Representative microscopic images of cells from the abdominal cavity 1 hr after i.a. injection of either ATP-loaded or empty OMV. OMV: DiI, Nucleus: DAPI, Neutrophils: Ly-6G-FITC. (**C**) Cells from remote organs were isolated 1 hr after i.a. injection of either ATP-loaded or empty OMV. OMV were mainly taken up by neutrophils (except in the spleen, ratio ≈ 1). t-Test with Benjamini-Hochberg correction, n=5 animals per group of N=2 independent experiments. Means and individual values are shown. (**D**) Representative microscopic image of pulmonary neutrophils 1 hr after i.a. injection of either ATP-loaded or empty OMV. OMV co-localize with the endolysosomal compartment. OMV: DiI, Endolysosomal system: LysoTracker Deep Red, Neutrophils: Ly-6G-FITC. (**E**) Pulmonary neutrophils were isolated 1 hr after i.a. injection of ATPγs-loaded or empty OMV, bead-sorted, and RNA sequencing was done. Principal component analysis shows significantly different clustering between neutrophils that took up ATPγs-loaded (NA) or empty OMV (NE). PERMANOVA, n=6 animals in the NE group, n=5 animals in the NA group. Ellipses represent 95% confidence level. (**F**) Volcano plot of RNA sequencing results shows an upregulation of genes mainly in the NA group. Genes classified in either lysosome (LYSO) or neutrophil degranulation pathways (NDG) or both, which were mentioned in the text, were highlighted. (**G**) Heatmap of the lysosome pathway (LYSO) showing the gene expression per sample. (**H**) Heatmap of the neutrophil degranulation pathway (NDG) showing the gene expression per sample.

The online version of this article includes the following figure supplement(s) for figure 6:

**Figure supplement 1.** Uptake of outer membrane vesicles (OMV) by neutrophils.

**Figure supplement 2.** Characterization of local immune response in the abdominal cavity.

*Figure 6 continued on next page*

*Figure 6 continued*

**Figure supplement 3.** Assessment of outer membrane vesicle (OMV) uptake by immune cells in remote organs.

**Figure supplement 4.** Assessment of the purity of bead-sorted pulmonary neutrophils.

**Figure supplement 5.** List of significantly different pathways after enrichment analysis of RNA sequencing results.

Within 1 hr, the DiI-stained OMV were distributed throughout the body and mainly taken up by neutrophils, with remarkable differences between organs (*Figure 6—figure supplement 3A and B, C*). The highest ratio of OMV$^{pos}$ neutrophils compared to all other OMV$^{pos}$ cells was observed in the lung in an ATP-independent manner (*Figure 6C*). Intracellular fate of engulfed OMV is still a matter of discussion but one possibility is the degradation in the endolysosomal compartment (*Bielaszewska et al., 2017*; *Juodeikis and Carding, 2022*). We therefore injected DiI-stained OMV i.a. to assess to which membrane compartment OMV co-localize. A strong co-localization with the endolysosomal compartment of pulmonary neutrophils was identified (*Figure 6D*).

To determine what pathways OMV-derived bacterial ATP initiates in neutrophils after uptake into the endolysosomal compartment, pulmonary neutrophils were isolated (*Figure 6—figure supplement 4*) for RNA sequencing. Uptake of ATP-loaded OMV resulted in a significantly different transcriptional activity compared to empty OMV (*Figure 6E*): The strongest differences were found in lysosomal and neutrophil degranulation pathways (*Figure 6F*, *Figure 6—figure supplement 5*). In the lysosomal pathway, transcripts for proteins involved in autophagosome-lysosome fusion (Gabarap), endosome to lysosome transport (Hgsnat, Laptm5), or enzymatic cleavage (Capn1, Fuca1, Ctsb, Ctsd, Gba) were significantly higher expressed in response to ATP-loaded OMV when compared to empty OMV (*Figure 6F and G*). Furthermore, neutrophil degranulation was significantly increased in response to ATP-loaded OMV as indicated by increased expression of gene transcripts resulting in antibacterial activity (Lyz2, S100a8, Camp) or NETosis (Tlr2, Itgb2) (*Figure 6F and H*).

In summary, OMV are phagocytosed locally by major cell populations with phagocytic ability. Remotely, OMV were mainly taken up by neutrophils, where they co-localize with the endolysosomal compartment. Delivery of ATP by OMV resulted in an upregulation of lysosomal activation and neutrophil degranulation.

## Discussion

In this study, we have demonstrated that ATP release is dependent on the respiratory chain located at the inner bacterial membrane. ATP synthase is most likely dominant compared to cytochrome $bo_3$ oxidase because of its non-redundant function for ATP generation and bacterial growth. Mutations in one of the cytochrome $bo_3$ oxidase subunits cyoA, cyoC, or cyoD could be compensated by another subunit. This is supported by lower ATP release in the cyoB mutant, which harbors all three prosthetic groups of the cytochrome $bo_3$ oxidase and is therefore indispensable for proper function (*Tsubaki et al., 2000*). Alternatively, deficiency in the $bo_3$-type oxidase could be partially compensated by $bd$-type oxidases (*Borisov et al., 2011*; *Grauel et al., 2021*).

Instability of the outer bacterial membrane as we have seen for the ΔompF mutant (*Choi and Lee, 2019*) is associated with bacterial death and ATP release. It remains unclear whether loss of ATP results directly in bacterial death or bacterial death is the direct result of impaired outer bacterial membrane stability and ATP release is secondary to bacterial death. However, since the inner membrane and ATP generation remains intact and the periplasmic space is considered generally devoid of ATP, the latter seems more likely (*Kulp and Kuehn, 2010*). Previous studies used Sytox/PI staining and microscopy to identify bacterial death and did not identify bacterial death as a relevant source of extracellular ATP during exponential growth (*Hironaka et al., 2013*). Conversely, we used flow cytometry, a quantitatively more sensitive method, and our data demonstrated that impaired membrane integrity and bacterial death are critical for ATP release during growth. Despite low rates of bacterial death during exponential growth (*Koch, 1959*), and a low ratio of dead to live bacteria, the high gradient between intra- and extracellular ATP (approximately 100–1000× higher) might be sufficient to explain the amount of released ATP measured (*Leduc and van Heijenoort, 1980*; *Mempin et al., 2013*).

ATP was detected within OMV and the concentration of ATP within OMV is sufficient to activate P2-type receptors in lysosomes (*Araujo et al., 2020*; *Junger, 2011*). Furthermore, OMV are released in response to environmental stresses like infection and sepsis (*Kulp and Kuehn, 2010*;

*Orench-Rivera and Kuehn, 2016*). ATP is likely to activate P2 receptors, which have been shown to be expressed in lysosomes, such as P2X1, P2X3, or P2X4 (*Qureshi et al., 2007*; *Robinson and Murrell-Lagnado, 2013*). OMV can therefore be considered as non-living bacteria-resembling highly inflammatory vesicles (*Jang et al., 2015*; *Kulp and Kuehn, 2010*) that effectively distribute ATP throughout the body to activate cells, such as neutrophils, initiating systemic inflammatory responses. Activation of neutrophils in response to ATP-loaded OMV resulted in upregulation of inflammatory, lysosomal, and neutrophil degranulation pathways as well as an upregulation of the apoptosis pathway and genes involved in NETosis by a hitherto unknown mechanism. These pathways explain in part the low neutrophil counts we observed in mice. Such remote degranulation in response to OMV is unlikely to control bacteria at the source of sepsis but rather harmful to host tissues increasing sepsis severity (*Eichelberger and Goldman, 2020*).

The study has several limitations: Bacterial ATP release is strain specific (*Figure 1*; *Mempin et al., 2013*). As we focused on the laboratory BW25113 *E. coli* strain, it remains to be elucidated if impaired outer membrane integrity and bacterial death are also of such importance in other gram$^{neg}$ or also gram$^{pos}$ bacteria. The approach to load OMV with ATP is rather artificial; it is, however, the only way to assure that ATP-loaded and empty OMV only differ in their ATP cargo. OMV have surface antigens and contain DNA, proteins, and other metabolites (*Kulp and Kuehn, 2010*), which are known to elicit inflammation as well. This was controlled using the same OMV as baseline vehicles.

Several open questions need to be addressed in future projects: It remains to be determined how OMV are physiologically loaded with ATP. Potentially, despite the lack of a transporter, ATP may similarly to eukaryotic cells leak (*Yegutkin et al., 2006*) across the inner membrane into the periplasmic space that lacks the enzymes for ATP generation. Different types of OMV have been described in recent years, e.g., outer-inner membrane vesicles (O-IMV) or explosive OMV (*Juodeikis and Carding, 2022*; *Takaki et al., 2020*; *Toyofuku et al., 2019*; *Turnbull et al., 2016*; *Turner et al., 2015*), which are composed of outer membrane, inner membrane, and cytoplasmic content. The mechanisms, how these types of OMV are generated, could explain that ATP is found within them. However, the $\Delta tolB$ mutant produces, unlike other tol mutants, only very few O-IMV (0.1–2% of all OMV) (*Pérez-Cruz et al., 2013*; *Reimer et al., 2021*; *Takaki et al., 2020*) and our western blot analysis suggests that our OMV are primarily outer membrane derived. Future studies may address the ATP cargo of the different OMV subgroups.

OMV are promising diagnostic molecular biomarkers in gram$^{neg}$ sepsis (*Michel and Gaborski, 2022*) and it would be of relevance to compare OMV isolated from septic and control patients to assess possible differences in ATP cargo. Furthermore, it remains to be elucidated, which mechanisms lead to low local neutrophil counts. Since ATP is involved in cell death (*Proietti et al., 2019*) as well as in chemotaxis (*Junger, 2011*), increased cell death, impaired infiltration, or a combination of both is possible.

This study reveals that ATP is released by bacteria during growth because of impaired membrane integrity and bacterial death. ATP is also being released via OMV and therefore acts locally (direct release) and systemically (via OMV). Bacterial ATP reduces neutrophil counts, activates the endolysosomal system, and upregulates neutrophil degranulation, which together increase the severity of abdominal infection and early sepsis. These findings have the potential to lead to the development of novel treatments for abdominal sepsis, e.g., by in vitro generated OMV that modulate neutrophil function via delivery of inhibitors to intracellular purinergic receptors during sepsis.

## Materials and methods
### Human data

From five patients that underwent revision laparotomy because of abdominal sepsis, swabs were taken from abdominal fluid, streaked on LB agar plates (15 g Agar, 5 g Bacto Yeast Extract, 10 g Bacto Tryptone, 5 g NaCl in 1 l ddH$_2$O; Key resources table) and cultivated an/aerobically for 48 hr. The human experimental protocol was approved by the Cantonal Ethics Commission Bern, Switzerland (ethical approval 2017-00573, NCT03554148). Written informed consent was obtained from all patients and the study has been performed in accordance with the Declaration of Helsinki as well as the CONSORT statement.

## Mouse handling

Specific pathogen-free C57Bl/6JRccHsd mice (Key resources table) were purchased at the age of 8 weeks from Inotiv (earlier Envigo, the Netherlands) and were housed in ventilated cages in the Central Animal Facility, University of Bern, Switzerland. All experiments were performed in the morning, mice were supplied with a 12 hr light/dark cycle at 22°C and fed ad libitum with chow and water. To mini-mize cage effects, we mixed mice over several cages and therefore only used female mice. All animal procedures were carried out in accordance with the Swiss guidelines for the care and use of laboratory animals as well as in accordance with the ARRIVE guidelines and were approved by the Animal Care Committee of the Canton of Bern (Switzerland) under the following number: BE41/2022.

## CLP sepsis model

To isolate bacteria from mice with abdominal sepsis, CLP was performed as described elsewhere with some minor modifications (*Rittirsch et al., 2009*). In brief, mice were anesthetized s.c. injecting (3 µl/g body weight) a mixture of fentanyl (0.05 mg/ml), dormicum (5 mg/ml), and medetor (1 mg/ml) and were then shaved and disinfected with Betadine. Mid-line laparotomy was performed (approximately 1 cm) and the cecum was exposed. The proximal one third of the cecum was ligated with Vicryl 4-0 (Ethicon, cat# V1224H) and perforated with a 23 G needle. The cecum was returned to the abdom-inal cavity and the laparotomy was sutured continuously in two layers with prolene 6-0 (Ethicon, cat# MPP8697H). At the end, the antidote (naloxone [0.1 mg/ml], revertor [5 mg/ml], temgesic [0.3 mg/ml]) was s.c. injected (9 µl/g body weight). A semiquantitative score sheet was used to predict animal postoperative well-being. Mice were evaluated every 4 hr according to the following criteria: appear-ance, level of consciousness, activity, response to stimulus, eye shape, respiratory rate, and respiratory quality and analgesia was applied if necessary. After 10 hr, abdominal fluid was collected, spread on LB agar plates, and cultivated an/aerobically for 48 hr.

## Whole 16S-rRNA Sanger sequencing

Twenty-five colonies cultivated from abdominal fluid of patients and mice each described above were randomly picked and collected in separate sterile Eppendorf tubes and resuspended in 1 ml of sterile PBS. Each sample was centrifuged for 5 min at 20,000×$g$ and washed once with 1 ml sterile PBS. The pellet was resuspended in 20 µl of sterile PBS and samples were incubated 5 min at 100°C. 1 µl was used for PCR using GoTaq G2 Green Master Mix (Key resources table) and the following primers (Key resources table) at a final concentration of 0.2 µM:

-fD1: 5'-AGA-GTT-TGA-TCC-TGG-CTC-AG-3'
-fD2: 5'-AGA-GTT-TGA-TCA-TGG-CTC-AG-3'
-rP1: 5'-ACG-GTT-ACC-TTG-TTA-CGA-CTT-3'

PCR conditions were as follows: Initial 5 min at 94°C for denaturation, followed by 35 cycles of 1 min denaturation at 94°C, 1 min annealing at 43°C, and 2 min extension at 72°C, with a final exten-sion for 10 min at 72°C. This resulted in a PCR product of optimally ~1400–1500 bp. 20 µl PCR product were run on 2% agarose gel for 90 min, cut out and purified using QIAquick Gel Extraction Kit (Key resources table). The amplicon concentration was measured using nanodrop (Thermo Fisher Scientific) and Sanger sequencing was done by Microsynth.

## Quantification of released bacterial ATP

Bacteria were aerobically grown in 20 ml LB medium (5 g Bacto Yeast Extract, 10 g Bacto Tryptone, 5 g NaCl in 1 l ddH$_2$O; Key resources table) overnight (O/N) (16 hr/37°C/200 rpm) and 0.25 ml of O/N culture were diluted in 100 ml fresh LB medium. Up to 24 hr, 0.25 ml bacteria culture were taken at several time points for growth assessment (OD$_{600}$) and 1 ml was taken for ATP quantification. OD$_{600}$ was measured using a Tecan Spark spectrophotometer. The 1 ml sample for ATP quantification was centrifuged for 5 min at 16,000×$g$, the supernatant was filtered through a 0.2 µm syringe filter and stored at –80°C. ATP was quantified using a luciferin-luciferase-based assay according to the manu-facturer's protocol (ATP Kit SL, Key resources table) and bioluminescence was measured using a Tecan Spark spectrophotometer. In *Figure 3*, LB medium was supplemented with 1 mM calcium or 0.5 mM EDTA for the controls.

## Absolute quantification of bacteria and assessment of viability

Growth culture was set up as described above and after 4 hr (ATP peak), samples were taken to quantify bacteria and assess viability. Bacteria were diluted in PBS and stained using the Cell Viability Kit with BD Liquid Counting Beads (Key resources table) according to the manufacturer's protocol. In brief, 100 µl of diluted bacterial growth culture was stained with 1 µl TO dye, 1 µl PI dye, and 10 µl of counting beads were added. The sample was acquired on a CytoFlex S (Beckman Coulter), setting the thresholds for the TO- and the PI-channel to 1000 and analysis was done with FlowJo software (Key resources table). Bacterial biomass was calculated to bacteria/ml culture according to the manufacturer's protocol.

## Transformation of *E. coli* PS

The Keio collection (Key resources table) *E. coli* PS was aerobically grown in LB medium O/N (16 hr/37°C/200 rpm). 0.5 ml O/N culture was diluted in 35 ml fresh LB medium and incubated until $OD_{600}$ was 0.35–0.45. Cultures were chilled in ice-water and washed two times with ice-cold 20 ml ultrapure water (Thermo Fisher Scientific, cat# 10977035) (20 min/4°C/3200×$g$). The supernatant was carefully decanted, and the bacterial pellet was gently resuspended by pipetting (no vortex). After the final wash, the pellet was resuspended in 240 µl ice-cold ultrapure water and kept on ice until EP. 80 µl of ice-cold bacterial cells were mixed with 1 µl plasmid (pBAD28==pEMPTY or pHND10==pAPY, 100–200 ng, Key resources table) in chilled 0.1 cm cuvettes (Bio-Rad, cat# 1652089) and immediately electroporated (voltage = 1.8 kV, capacitance = 25 µF, resistance = 200 Ω). Directly after EP, 1 ml of warm LB medium was added to the cuvette, bacteria were transferred to a tube containing 10 ml pre-warmed LB medium and cultivated for 1 hr at 37°C. Then, bacteria were dispersed on LB agar plates supplemented with ampicillin (100 µg/ml) and incubated at 37°C. The next day, three colonies were picked, streaked on a new ampicillin supplemented LB agar plate, and incubated for 24 hr. The following day, two cryostocks were made from single colonies in LB medium supplemented with 20% glycerol.

## Bacteria injection sepsis model

The Keio collection (Key resources table) *E. coli* PS transformed with either pEMPTY or pAPY was aerobically grown O/N (16 hr/37°C/200 rpm) in 20 ml LB medium supplemented with ampicillin (100 µg/ml). PS+pEMPTY and PS+pAPY were washed with PBS (20 min/22°C/3200×$g$) and resuspended in 20 ml fresh LB medium supplemented with ampicillin (100 µg/ml). To induce the apyrase, L-arabinose (0.03%, Sigma-Aldrich, cat# A3256-25G) was added. After 3 hr, bacteria were washed, resuspended in PBS supplemented with L-arabinose (0.03%) and 2×10$^9$ colony forming units were injected i.a. ATP in bacteria cultures and in the abdominal fluid were assessed using the same assay as described above (ATP Kit SL, Key resources table). Mice were evaluated for postoperative well-being every 4 hr as described above. After 4 and 8 hr and when the score reached specific criteria (for the survival experiment), animals were sacrificed using pentobarbital (150 mg/kg body weight) followed by organ collection.

## Collection of abdominal fluid, blood, and organs

After pentobarbital injection, the mice were placed on a surgical tray, fixed, the abdomen was shaved, the abdominal skin was disinfected with Betadine and the skin (but not peritoneum) was cut. Abdominal cells were isolated as described elsewhere with some modifications (*Ray and Dittel, 2010*). In brief, a 22 G peripheral IV catheter was inserted into the abdominal cavity through the peritoneum. The abdominal cavity was flushed two times with 5 ml MACS buffer (PBS supplemented with 3% FBS [Gibco, cat# 10500-064], 2% HEPES [Sigma-Aldrich, cat# H0887-100ml], and 2 mM EDTA [Sigma-Aldrich, cat# E5134-500G]). The first 5 ml were used to flush the upper abdomen under pressure, and the second 5 ml, to flush the lower abdomen under pressure. Part of the aspirated fluid was used for aerobic plating, if needed, and the rest was centrifuged for 5 min at 700×$g$ to pellet the abdominal cells for various downstream applications. To collect blood, the peritoneum was opened. 300 µl blood was collected from the inferior vena cava using a 22 G peripheral IV catheter and a 1 ml syringe, which was prefilled with 30 µl 2 mM EDTA. Before organ collection (lungs, liver, kidney, and spleen), the mouse was flushed with 5 ml PBS. Organs were excised and used for downstream processing.

## Digestion of organs and preparation of single-cell suspension

Harvested organs were digested (Key resources table; kidney and lungs: 1 mg/ml Col I+1 mg/ml Col IV+1 mg/ml Col D+0.1 mg/ml DNase I in DMEM [Gibco, cat# 31966-047]+3% FBS for 30 min; liver: 1 mg/ml Col IV+0.1 mg/ml DNase I in DMEM+3% FBS for 20 min) at 37°C with a spinning magnet or gently pushed through a 100 μm mesh (spleen). After washing with MACS buffer (5 min/4°C/700×$g$), erythrocytes were lysed using self-made RBC buffer (90 g $NH_4Cl$, 10 g $KHCO_3$, 370 mg EDTA in 1 l ddH$_2$O for 10× stock). Cells were washed again and stained as described below.

## Staining of cells and flow cytometry

First, viability dye (Key resources table) and Fc-block (Key resources table) were diluted in PBS and cells were incubated for 20 min at 4°C in the dark. Cells were washed with MACS buffer and surface staining was done with the listed antibody cocktail:

> (Key resources table: Ly-6G FITC, Ly-6C PerCP-Cy5.5 or Ly-6C PE-Cy7, CD11b APC, CD206 AF700, CD11c APC-eFluor780, CD45 efluor450 or CD45 APC-Cy7, CD19 Super Bright 600, CD3 BV605, NK1.1 BV605, CCR2 BV650, I-A/I-E BV711, CX3CR1 BV785, Siglec F PE, FcεR1α PE/Dazzle 594, CD115 PE-Cy7, F4/80 BUV395)

for 20 min at 4°C in the dark. Cells were washed again and resuspended in MACS buffer for acquisition on an LSR-Fortessa (BD Biosciences). Analysis was done with FlowJo software (Key resources table) and OMIQ web-based analysis platform (https://www.omiq.ai/). To preserve the global structure of abdominal cell populations, uniform manifold approximation and projection was used as dimensionality reduction technique and cell populations were defined using FlowSOM clustering algorithm (*Van Gassen et al., 2015*).

## Absolute quantification of bacteria by plating

To count bacteria in isolated abdominal fluid, blood, or growth cultures, serial dilutions were done (1 to 1:100,000) and 50 μl of each dilution was streaked on LB agar plates and aerobically incubated. Plates which had between 20 and 200 colony forming units were used for quantification.

## Collection of OMV and ATP measurement of OMV

*E. coli* PS or hypervesiculation mutants from the Keio collection were grown in LB medium for 5 hr, O/N (16 hr) or 24 hr at 37°C and 200 rpm. Bacteria cultures were then centrifuged (20 min/4°C/3200×$g$) to pellet bacteria. The supernatant was filtered through a 0.45 μm PES filter (Key resources table) and ultracentrifuged (1.5 hr/4°C/150,000×$g$) to pellet OMV. When ATP within OMV was measured, OMV were washed in PBS and directly stored at –80°C. ATP was quantified using a luciferin-luciferase-based assay according to the manufacturer's protocol (Microbial ATP Kit HS, Key resources table) and bioluminescence was measured using Tecan Spark spectrophotometer.

If OMV were used for characterization or i.a. injection, first ultracentrifugation was followed by a density gradient ultracentrifugation using OptiPrep (Iodixanol, STEMCELL Technologies, cat# 07820). OMV were resuspended in 50% OptiPrep and OptiPrep gradient (10%, 20%, 30%, 40%, 45%, 50%) was made in underlay technique starting with the 10% layer. Samples were ultracentrifuged (16 hr/4°C/150,000×$g$) and six fractions (fractions 1, 2, 3a, 3b, 4, and 5) were defined. Fractions were washed separately with PBS (1.5 hr/4°C/150,000×$g$) and the amount of OMV was determined measuring protein concentration using BCA assay according to the manufacturer's protocol (Thermo Fisher Scientific, cat# 23227). For experiments, only OMV from fraction 3 were used, which were washed with PBS (1.5 hr/4°C/150,000×$g$), resuspended in PBS, and stored at –80°C until further processing.

## EP and staining of OMV

If EP and staining of OMV was performed, OMV were thawed, pelleted (1.5 hr/4°C/150,000×$g$), the pellet was resuspended in 720 μl EP buffer (500 mM sucrose and 1 ml glycerol in 10 ml ultrapure water) and kept on ice. This suspension was mixed with either 80 μl EP buffer (control) or 80 μl 10 mM ATP (or ATPγs for RNA sequencing experiment) (Key resources table), filled in chilled 0.4 mm EP cuvettes (Bio-Rad, cat# 1652088) and immediately electroporated (voltage = 1100 V, capacitance = 50 μF, resistance = 100 Ω). After EP, the OMV were 1:1 diluted in warm PBS and kept at 37°C for

20 min. DiI (Key resources table) was added 1:100 to the sample during the incubation time. OMV suspension was then washed with PBS (1.5 hr/4°C/150,000×*g*).

## OMV injection sepsis model

After the final wash in PBS (see above), OMV were resuspended in NaCl 0.9% and filtered through a 0.45 µm centrifuge filter tube (Sigma-Aldrich, cat# CLS8162-96EA). The final OMV suspension was quantified using a BCA assay. 25 µg of either ATP-loaded OMV or empty OMV were i.a. injected and mice were evaluated as described above. After 1 hr, the animal was sacrificed using pentobarbital followed by organ collection. Tissue digestion and flow cytometry was done as described above.

## Assessment of ATP release by OMV

OMV were electroporated as described above. After washing in PBS (1.5 hr/4°C/150,000×*g*), the OMV pellet was resuspended in 6 ml warm PBS and incubated at 37°C. At baseline, after 1, 3, 5, 10, and 24 hr, 1 ml was taken and washed with PBS (1.5 hr/4°C, 150,000×*g*). The OMV pellet was then resuspended in 200 µl PBS to assess protein concentration using a BCA assay, and ATP in OMV was quantified using a luciferin-luciferase-based assay according to the manufacturer's protocol (Intracellular ATP Kit HS, Key resources table). Bioluminescence was measured using a Tecan Spark spectrophotometer.

## SDS-PAGE, protein staining, and western blot

20 µl of diluted *E. coli* culture and 20 µl of the different OMV fractions (except fraction 1, since no protein could be detected) were diluted 1:1 with Laemmli/βME (Laemmli buffer solution containing 5% β-mercaptoethanol). The mixture was heated at 100°C for 5 min, shortly centrifuged at 13,000 rpm and loaded on Mini-PROTEAN TGX Gels (Bio-Rad, cat# 4561094). Bio-Rad marker (6 µl) was added to one well and the gel was run at 100 V for 1.5 hr. The gel was then either directly stained with Coomassie blue or transferred to a membrane.

Coomassie blue staining was done as follows: The gel was washed in ddH$_2$O and the staining solution (Coomassie blue 0.1%, 40% ethanol, 10% acetic acid) was heated for 15 s in the microwave. Warm staining solution was added to the gel and gently shaken for 15 min. Then, staining solution was removed, and the gel was washed with ddH$_2$O. Destaining solution (10% ethanol, 7.5% acetic acid) was added for 1 hr, exchanged and left O/N.

To transfer the proteins to a membrane, iBlot2 (Invitrogen) was used. Membranes were then blocked with 4% milk/PBS for 1 hr. FtsZ-antibody (1:200, Key resources table) and ompF-antibody (1:500, Key resources table) were added and incubated O/N at 4°C. The next day, membranes were washed with PBS Tween (PBST, 0.05%) three times for 5 min. Secondary fluorescent antibody was then added in milk (1:10,000, Key resources table) and membrane was incubated for 1 hr. The membrane was washed again with PBST three times for 5 min and then scanned using a Licor Odyssey.

## Nanoparticle tracking analysis

Size distribution of OMV was analyzed using the NanoSight NS300 Instrument (Malvern Panalytical, 405 nm laser) according to the manufacturer's protocols. OMV were resuspended in PBS and serial dilutions (1 to 1:100,000) were used to find suitable concentrations. Each experimental sample was analyzed five times. PBS was used to flush the system between the samples and to assess background. For each sample, the relative amount of OMV and the OMV size was recorded, which resulted in a size distribution curve. NTA 2.3 software was used to analyze the data and the following script was used for acquisition (*Gheinani et al., 2018*): SETTEMP 25; CAMERAON; CAMERAGAIN 12; CAMERALEVEL 11; REPEATSTART; SYRINGLOAD 100; DELAY 10; SYRINGSTOP; DELAY 15; CAPTURE 60; DELAY 1; **REPEAT 4**; SETTEMP OFF; PROCESSINGLESETTING; EXPORTRESULTS.

## Electron microscopy negative staining

For imaging of negatively stained samples, 5 µl of OMV suspension were adsorbed on glow discharged carbon-coated 400 mesh copper grids (Plano) for 1–5 min. After washing the grids three times by dipping in ultrapure water, the grids were stained with 2% uranyl acetate solution (Electron Microscopy Science) in water for 45 s. The excess fluid was removed by gently pushing the grids sideways onto filter paper. The grids were then examined with an FEI Tecnai Spirit transmission electron microscope at 80 kV, which was equipped with a Veleta TEM CCD camera (Olympus).

## Immunofluorescent microscopy

One hour after i.a. injection of DiI-stained OMV, lungs were digested as described above. Single-cell suspension was either fixed and imaged or live cells were imaged. For the fixed approach, single-cell suspension was immunolabeled with FITC-tagged anti-Ly-6G (Key resources table) at 1:100 dilution for 20 min at 4°C in the dark to distinguish neutrophils. Cells were washed once with PBS and applied onto glass slides using Cytospin (Thermo Shandon) for 5 min (1800 rpm). Slides were air-dried for 2 min, washed with PBS, and fixed using 4% paraformaldehyde solution for 10 min. Cell nuclei were stained with DAPI (Key resources table) at 1:5000 concentration in IF buffer (0.25% BSA, 0.1% Triton X-100 in PBS) for 1.5 hr at room temperature in the dark. After washing 3×5 min with IF buffer, the slides were covered and sealed with nail polish.

If live cells were imaged, single-cell suspension was concurrently immunolabeled and stained with Hoechst 33342 (Key resources table) at 1:1000 dilution for cell nuclei, FITC-tagged anti-Ly-6G (Key resources table) at 1:100 dilution for neutrophils and LysoTracker Deep Red (Key resources table) at 1:1000 dilution for lysosomes for 30 min at 37°C in the dark. Cells were washed with MACS buffer and immediately imaged.

Fluorescence images were taken using either a Zeiss LSM710 or a Zeiss LSM980 inverted confocal laser scanning microscope equipped with a ×63 (NA1.4) oil immersion objective. Detector wavelength cutoffs were set to minimize signal crosstalk between fluorophores.

## RNA isolation from pulmonary neutrophils

One hour after OMV i.a. injection, lungs were digested as described above. One animal had to be excluded, since the surgical time point has been missed. Neutrophils were isolated from single-cell suspension using Streptavidin MicroBeads (Miltenyi Biotec, cat# 130-048-101) according to the manufacturer's protocol. In brief, cells were counted and incubated with a biotinylated anti-mouse Ly-6G antibody (Key resources table) for 20 min at 4°C. After washing, Streptavidin MicroBeads were added and incubated for 20 min at 4°C. After washing, a MidiMACS separator (Miltenyi Biotec, cat# 130-042-302) together with an LS column (Miltenyi Biotec, cat# 130-042-401) was used to isolate neutrophils and purity of positively selected neutrophils as well as unlabeled cells was assessed by flow cytometry (Key resources table: Ly-6C PE-Cy7, CD11b APC). Neutrophils were pelleted, and supernatant was removed by pipetting. Neutrophils were snap-frozen and stored at –80°C. RNA was isolated directly from frozen neutrophils pellets using Promega ReliaPrep RNA Cell Miniprep System (Key resources table) according to the manufacturer's protocol and quality was assessed using Bioanalyzer and RNA 6000 Nano Kit (RQN 5.8–9.6, median 7.55). Samples were snap-frozen and stored at –80°C until further processing by the Next Generation Sequencing Platform, University of Bern, https://www.ngs.unibe.ch/.

## RNA sequencing

RNA sequencing libraries were prepared using Lexogen CORALL total RNA-seq library kit according to the manufacturer's protocol, which includes an rRNA depletion step. Sequencing was performed on an Illumina NovaSeq6000 SP flow cell, 2×50 cycles.

### Alignment and quantification

The resulting fastq files were quality controlled using fastqc v0.11.9 (*Babraham Bioinformatics, 2024*). Forward R1 reads were debarcoded by moving the first 12 nucleotides on the 5' end to the name of the read via fastp v0.19.5 (*Chen et al., 2018*). The alignment was performed with STAR v2.7.10a_alpha_220818 (*Dobin et al., 2013*). First, a genome index was generated from the mouse reference genome GRCm39 ENSEMBL v108. Second, the reads were then aligned to the reference using STAR with default parameters. Gene read counts were quantified using featureCounts from subread v2.0.1 using the mm108 GTF annotation with default parameters (*Liao et al., 2014*).

### Data visualization

The data from the read count matrix was normalized to reads per million and log-transformed, $x \rightarrow \log(1+x)$. The resulting data was used for principal component analysis, which was performed with the R function *prcomp* and visualized with a custom script using ggplot2 (*Wickham, 2016*).

### Differential gene expression

Differentially expressed genes were computed using the R package DESeq2 (*Love et al., 2014*).

### Pathway enrichment analysis

The differentially expressed genes obtained from DESeq2 with an adjusted p-value below 0.01 were uploaded to Metascape for pathway analysis on December 5, 2023 (*Zhou et al., 2019*).

### Heatmaps

The lists of genes of the pathways of interest were obtained from genome.jp for the KEGG pathways and using the R function *gconvert* from the R package gprofiler2 otherwise (*Kolberg and Raudvere, 2023*). The log-transformed data was used and the heatmaps were done with the heatmap.2 function of the R package gplots with the hierarchical cluster method 'complete' using Pearson's correlation distance (*Warnes et al., 2022*).

### Volcano plot

The volcano plots were performed with the R package ggplot2 and a custom R script (*Wickham, 2016*).

## Statistical analysis

Descriptive statistics and statistical tests were performed using Prism software (Key resources table) or R and RStudio (Key resources table). For differences between two groups a t-test was applied when data was normally distributed (parametric). Otherwise, a Wilcoxon rank-sum test was used (nonparametric). For differences between more than two groups, a one-way ANOVA followed by a Tukey post hoc test (parametric), or a Kruskal-Wallis test followed by a pairwise Wilcoxon rank-sum test with Benjamini-Hochberg correction (nonparametric) was used. For survival analyses, a log-rank test was used. No one-tailed tests were used. No method was used to predetermine experimental sample sizes. A linear model and the correlation between ATP and $OD_{600}$ in *Figure 2* and *Figure 3* were computed using the R functions *lm* and *cor.test* with the following parameters:

stats::cor.test(*data*$*variable*ATP, *data*$*variable*growth, alternative = "two.sided", method = "pearson") summary(stats::lm(*variable*ATP ~ *variable*growth, data = *data*)).

p<0.05 (*P*==padj, when correction for multiple testing was necessary) was considered significant in all statistical analyses unless stated otherwise in the figure legend. Significant differences are in *italic*.

## Acknowledgements

We are grateful for the technical and financial support of the Inselspital, the DBMR, and our laboratory. A special thanks goes to Dana Leuenberger and Isabel Büchi for their support with the animal experiments and microscopy. We also want to thank the Microscopy Imaging Center (MIC) of the University of Bern for their assistance in sample preparation and imaging on the electron microscope. We want to thank Prof. Dr. Fabio Grassi and his group for providing us with the pBAD28 and pHND10 plasmids. We also want to thank Prof. Dr. Siegfried Hapfelmeier, Dr. Olivier Schären, and Prof. Dr. Torsten Seuberlich for their extremely helpful scientific support. We want to thank Prof. Dr. Andrew Macpherson for supporting us with his expertise and providing us with the Keio collection. Finally, we want to thank Prof. Dr. Joel Zindel and Dr. Felix Baier for their critical contributions and ideas to complete this project. Figures were created using BioRender.com.

This work was supported by the Swiss National Science Foundation (No.: 166594) to GB.

## Additional information

### Funding

| Funder | Grant reference number | Author |
|---|---|---|
| Swiss National Science Foundation | 166594 | Guido Beldi |

The funders had no role in study design, data collection and interpretation, or the decision to submit the work for publication.

### Author contributions

Daniel Spari, Conceptualization, Data curation, Software, Formal analysis, Validation, Investigation, Visualization, Methodology, Writing - original draft, Project administration, Writing - review and editing; Annina Schmid, Formal analysis, Investigation, Writing - original draft; Daniel Sanchez-Taltavull, Data curation, Software, Formal analysis, Visualization; Shaira Murugan, Investigation; Keely Keller, Data curation, Software, Writing - original draft; Nadia Ennaciri, Formal analysis, Investigation; Lilian Salm, Investigation, Methodology, Writing - original draft; Deborah Stroka, Resources, Funding acquisition, Methodology, Writing - original draft, Writing - review and editing; Guido Beldi, Conceptualization, Resources, Supervision, Funding acquisition, Methodology, Writing - original draft, Project administration, Writing - review and editing

### Author ORCIDs

Daniel Spari (ID) https://orcid.org/0000-0002-5028-0851
Lilian Salm (ID) https://orcid.org/0000-0002-6812-3678
Deborah Stroka (ID) https://orcid.org/0000-0002-3517-3871
Guido Beldi (ID) https://orcid.org/0000-0002-9914-3807

### Ethics

NCT03554148.

The human experimental protocol was approved by the Cantonal Ethics Commission Bern, Switzerland (ethical approval 2017-00573, NCT03554148). Written informed consent was obtained from all patients and the study has been performed in accordance with the Declaration of Helsinki as well as the CONSORT statement.

All animal procedures were carried out in accordance with the Swiss guidelines for the care and use of laboratory animals as well as in accordance with the ARRIVE guidelines and were approved by the Animal Care Committee of the Canton of Bern (Switzerland) under the following number: BE41/2022.

Reviewer #1 (Public review): https://doi.org/10.7554/eLife.96678.3.sa1
Reviewer #2 (Public review): https://doi.org/10.7554/eLife.96678.3.sa2
Author response https://doi.org/10.7554/eLife.96678.3.sa3

## Additional files

### Supplementary files
• MDAR checklist

### Data availability

RNA sequencing data has been deposited in Gene Expression Omnibus (GEO) under the accession code GSE272296. In addition, all raw data, metadata and code are publicly available at BORIS Portal of the University of Bern under a CC-BY licence: https://doi.org/10.48620/418.

The following datasets were generated:

| Author(s) | Year | Dataset title | Dataset URL | Database and Identifier |
|---|---|---|---|---|
| Spari D, Schmid A, Sánchez-Taltavull D, Murugan S, Keller K, Ennaciri N, Salm L, Stroka D, Beldi G | 2024 | Released Bacterial ATP Shapes Local and Systemic Inflammation during Abdominal Sepsis | https://www.ncbi.nlm.nih.gov/geo/query/acc.cgi?acc=GSE272296 | NCBI Gene Expression Omnibus, GSE272296 |
| Beldi G, Keller KA, Sánchez Taltavull D, Murugan S, Schmid AB, Spari D, Ennaciri NS, Salm L, Keogh-Stroka DM | 2024 | Released Bacterial ATP Shapes Local and Systemic Inflammation during Abdominal Sepsis | https://doi.org/10.48620/418 | BORIS Portal, 10.48620/418 |

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

# Appendix 1

## Appendix 1—key resources table

| Reagent type (species) or resource | Designation | Source or reference | Identifiers | Additional information |
|---|---|---|---|---|
| Strain, strain background (*Mus musculus, female*) | Wild-type mice | Inotiv, the Netherlands | C57Bl/6JRccHsd | |
| Strain, strain background (*Enterococcus faecalis*) | Iso1 | This paper | *NA* | See Materials and methods: Human data |
| Strain, strain background (*Escherichia coli*) | Iso2 | This paper | *NA* | See Materials and methods: Human data |
| Strain, strain background (*Klebsiella pneumoniae*) | Iso3 | This paper | *NA* | See Materials and methods: Human data |
| Strain, strain background (*Staphylococcus aureus*) | Iso4 | This paper | *NA* | See Materials and methods: Human data |
| Strain, strain background (*E. faecalis*) | Iso5 | This paper | *NA* | See Materials and methods: CLP sepsis model |
| Strain, strain background (*E. coli*) | Iso6 | This paper | *NA* | See Materials and methods: CLP sepsis model |
| Strain, strain background (*S. aureus*) | Iso7 | This paper | *NA* | See Materials and methods: CLP sepsis model |
| Strain, strain background (*E. coli*) | Parental strain (PS) | Keio collection; *Baba et al., 2006* | BW25113 | |
| Strain, strain background (*E. coli*) | ΔcyoA | Keio collection; *Baba et al., 2006* | JW0422-1 | See *Figure 2* |
| Strain, strain background (*E. coli*) | ΔcyoB | Keio collection; *Baba et al., 2006* | JW0421-1 | See *Figure 2* |
| Strain, strain background (*E. coli*) | ΔcyoC | Keio collection; *Baba et al., 2006* | JW0420-1 | See *Figure 2* |
| Strain, strain background (*E. coli*) | ΔcyoD | Keio collection; *Baba et al., 2006* | JW0419-1 | See *Figure 2* |
| Strain, strain background (*E. coli*) | ΔatpA | Keio collection; *Baba et al., 2006* | JW3712-1 | See *Figure 2* |
| Strain, strain background (*E. coli*) | ΔatpB | Keio collection; *Baba et al., 2006* | JW3716-1 | See *Figure 2* |
| Strain, strain background (*E. coli*) | ΔatpC | Keio collection; *Baba et al., 2006* | JW3709-2 | See *Figure 2* |
| Strain, strain background (*E. coli*) | ΔatpD | Keio collection; *Baba et al., 2006* | JW3710-1 | See *Figure 2* |
| Strain, strain background (*E. coli*) | ΔatpE | Keio collection; *Baba et al., 2006* | JW3715-1 | See *Figure 2* |
| Strain, strain background (*E. coli*) | ΔatpF | Keio collection; *Baba et al., 2006* | JW3714-2 | See *Figure 2* |
| Strain, strain background (*E. coli*) | ΔatpH | Keio collection; *Baba et al., 2006* | JW3713-1 | See *Figure 2* |
| Strain, strain background (*E. coli*) | ΔompF | Keio collection; *Baba et al., 2006* | JW0912-1 | See *Figure 3* |
| Strain, strain background (*E. coli*) | ΔompC | Keio collection; *Baba et al., 2006* | JW2203-1 | See *Figure 3* |

*Appendix 1 Continued on next page*

*Appendix 1 Continued*

| Reagent type (species) or resource | Designation | Source or reference | Identifiers | Additional information |
|---|---|---|---|---|
| Strain, strain background (*E. coli*) | Δ*lamB* | Keio collection; *Baba et al., 2006* | JW3996-1 | See *Figure 3* |
| Strain, strain background (*E. coli*) | Δ*phoE* | Keio collection; *Baba et al., 2006* | JW0231-1 | See *Figure 3* |
| Strain, strain background (*E. coli*) | Δ*mlaA* | Keio collection; *Baba et al., 2006* | JW2343-1 | See *Figure 5* |
| Strain, strain background (*E. coli*) | Δ*mlaE* | Keio collection; *Baba et al., 2006* | JW3161-1 | See *Figure 5* |
| Strain, strain background (*E. coli*) | Δ*nlpI* | Keio collection; *Baba et al., 2006* | JW3132-1 | See *Figure 5* |
| Strain, strain background (*E. coli*) | Δ*tolB* | Keio collection; *Baba et al., 2006* | JW5100-1 | See *Figure 5* |
| Strain, strain background (*E. coli*) | Δ*degP* | Keio collection; *Baba et al., 2006* | JW0157-1 | See *Figure 5* |
| Strain, strain background (*E. coli*) | Δ*rfaD* | Keio collection; *Baba et al., 2006* | JW3594-1 | See *Figure 5* |
| Strain, strain background (*E. coli*) | Δ*rodZ* | Keio collection; *Baba et al., 2006* | JW2500-1 | See *Figure 5* |
| Antibody | Purified anti-Ms CD16/32, monoclonal | BioLegend | Cat# 101302; clone 93; Lot# B298973; RRID: AB_312801 | (1:200) |
| Antibody | Rat anti-Ms Ly-6G (FITC), monoclonal | BD Biosciences | Cat# 551460; clone 1A8; Lot# 9068981; RRID: AB_394207 | (1:100) |
| Antibody | Rat anti-Ms Ly-6C (PerCP-Cyanine5.5), monoclonal | Thermo Fisher Scientific | Cat# 45-5932-82; clone HK1.4; Lot# 2309273; RRID: AB_2723343 | (1:100) |
| Antibody | Rat anti-Ms/Hs CD11b (APC), monoclonal | BioLegend | Cat# 101212; clone M1/70; Lot# B312600; RRID: AB_312795 | (1:800) |
| Antibody | Rat anti-Ms CD206 (AF700), monoclonal | BioLegend | Cat# 141733; clone C068C2; Lot# B278058; RRID: AB_2629636 | (1:300) |
| Antibody | Armenian hamster anti-Ms CD11c (APC-eFluor780), monoclonal | Thermo Fisher Scientific | Cat# 47-0114-80; clone N418; Lot# 2133269; RRID: AB_1548652 | (1:300) |
| Antibody | Rat anti-Ms CD45 (efluor450), monoclonal | Thermo Fisher Scientific | Cat# 48-0451-82; clone 30-F11; Lot# 2005853 RRID: AB_1518806 | (1:600) |
| Antibody | Rat anti-Ms CD19 (Super Bright 600), monoclonal | Thermo Fisher Scientific | Cat# 63-0193-82; clone eBio1D3; Lot# 2366423; RRID: AB_2637308 | (1:150) |
| Antibody | Rat anti-Ms CD3 (BV 605), monoclonal | BioLegend | Cat# 100237; clone 17A2; Lot# B389899; RRID: AB_2562039 | (1:100) |
| Antibody | Mouse anti-Ms NK1.1 (BV605), monoclonal | BioLegend | Cat# 108739; clone PK-136; Lot# B389899; RRID: AB_2562273 | (1:150) |
| Antibody | Rat anti-Ms CCR2 (BV650), monoclonal | BioLegend | Cat# 150613; clone SA203G11; Lot# B294599; RRID: AB_2721553 | (1:100) |
| Antibody | Rat anti-Ms I-A/I-E (BV711), monoclonal | BioLegend | Cat# 107643; clone M5/114.15.2; Lot# B299330; RRID: AB_2565976 | (1:600) |

*Appendix 1 Continued on next page*

*Appendix 1 Continued*

| Reagent type (species) or resource | Designation | Source or reference | Identifiers | Additional information |
|---|---|---|---|---|
| Antibody | Mouse anti-Ms CX3CR1 (BV785), monoclonal | BioLegend | Cat# 149029; clone SA011F11; Lot# B304744; RRID: AB_2565938 | (1:300) |
| Antibody | Rat anti-Ms Siglec F (PE), monoclonal | Thermo Fisher Scientific | Cat# 12-1702-80; clone 1RNM44N; Lot# 2252684; RRID: AB_2637129 | (1:300) |
| Antibody | Armenian hamster anti-Ms FcεR1α (PE/Dazzle 594), monoclonal | BioLegend | Cat# 134331; clone MAR-1; Lot# B280348; RRID: AB_2687240 | (1:300) |
| Antibody | Rat anti-Ms CD115 (PE-Cy7), monoclonal | BioLegend | Cat# 135523; clone AFS98; Lot# B268547; RRID: AB_2566459 | (1:600) |
| Antibody | Rat anti-Ms F4/80 (BUV395), monoclonal | BD Biosciences | Cat# 565614; clone T45-2342; Lot# 1104580; RRID: AB_2739304 | (1:150) |
| Antibody | Rat anti-Ms CD45 (APC-Cy7), monoclonal | BioLegend | Cat# 103115; clone 30-F11; Lot# *NA*; RRID: AB_312980 | (1:150) |
| Antibody | Rat anti-Ms Ly-6C (PE-Cy7), monoclonal | BioLegend | Cat# 128017; clone HK1.4; Lot# B331355; RRID: AB_1732093 | (1:600) |
| Antibody | Rat anti-Ms Ly-6G (Biotin), monoclonal | BioLegend | Cat# 127604; clone 1A8; Lot# B314606; RRID: AB_1186108 | (1:600) |
| Antibody | Rabbit anti-*E. coli* ftsZ, polyclonal | Agrisera | Cat# AS10715; RRID: AB_10754647 | (1:200) |
| Antibody | Rabbit anti-*E. coli* ompF, polyclonal | Biorbyt | Cat# orb308741; RRID: *NA* | (1:500) |
| Antibody | Goat anti-rabbit, polyclonal | LI-COR Biosciences | Cat# 925-68021; RRID: AB_2713919 | (1:10,000) |
| Recombinant DNA reagent | pBAD28==pEMPTY (plasmid) | *Proietti et al., 2019*; *Santapaola et al., 2006*; *Scribano et al., 2014* | Ampicillin resistance | |
| Recombinant DNA reagent | pHND10==pAPY (plasmid) | *Proietti et al., 2019*; *Santapaola et al., 2006*; *Scribano et al., 2014* | Ampicillin resistance | |
| Sequence-based reagent | fD1 | *Weisburg et al., 1991* | PCR primers | 5'-AGA-GTT-TGA-TCC-TGG-CTC-AG-3' |
| Sequence-based reagent | fD2 | *Weisburg et al., 1991* | PCR primers | 5'-AGA-GTT-TGA-TCA-TGG-CTC-AG-3' |
| Sequence-based reagent | rP1 | *Weisburg et al., 1991* | PCR primers | 5'-ACG-GTT-ACC-TTG-TTA-CGA-CTT-3' |
| Commercial assay or kit | QIAquick Gel Extraction Kit | QIAGEN | Cat# 28706 | |
| Commercial assay or kit | ATP Kit SL | BioThema | Cat# 144-041 | |
| Commercial assay or kit | Cell Viability Kit with BD Liquid Counting Beads | BD Biosciences | Cat# 349480 | |
| Commercial assay or kit | Microbial ATP Kit HS | BioThema | Cat# 266-112 | |
| Commercial assay or kit | Intracellular ATP Kit HS | BioThema | Cat# 266-111 | |
| Commercial assay or kit | ReliaPrep RNA Cell Miniprep System | Promega | Cat# Z6011 | |
| Chemical compound, drug | Bacto Yeast Extract | Gibco | Cat# 212750 | |
| Chemical compound, drug | Bacto Tryptone | Gibco | Cat# 211699 | |
| Chemical compound, drug | Agar | Sigma-Aldrich | Cat# 05039-500G | |

*Appendix 1 Continued on next page*

*Appendix 1 Continued*

| Reagent type (species) or resource | Designation | Source or reference | Identifiers | Additional information |
|---|---|---|---|---|
| Chemical compound, drug | GoTaq G2 Green Master Mix | Promega | Cat# M782A | |
| Chemical compound, drug | Collagenase I (Col I) | Sigma-Aldrich | Cat# C0130-100MG | |
| Chemical compound, drug | Collagenase IV (Col IV) | Worthington | Cat# LS004189 | |
| Chemical compound, drug | Collagenase D (Col D) | Roche | Cat# 11088858001 | |
| Chemical compound, drug | DNAse I | Roche | Cat# 63792800 | |
| Chemical compound, drug | Adenosine 5′-triphosphate disodium salt hydrate (ATP) | Sigma-Aldrich | Cat# 2383-1G | |
| Chemical compound, drug | Adenosine-5'-(γ-thio)-triphosphate, Tetralithium salt (ATPγs) | Jena Bioscience | Cat# NU-406-50 | |
| Software, algorithm | GraphPad Prism v9.5.1 | Prism GraphPad software | https://www.graphpad.com/ | |
| Software, algorithm | FlowJo v10.8.1 | FlowJo software | https://www.flowjo.com | |
| Software, algorithm | fastqc v0.11.9 | *Babraham Bioinformatics, 2024* | http://www.bioinformatics.babraham.ac.uk/projects/fastqc/ | |
| Software, algorithm | fastp v0.19.5 | *Chen et al., 2018* | https://github.com/OpenGene/fastp RRID:SCR_016962 | |
| Software, algorithm | STAR v2.7.10a_alpha_220818 | *Dobin et al., 2013* | https://github.com/alexdobin/STAR RRID:SCR_004463 | |
| Software, algorithm | subread v2.0.1 | *Liao et al., 2014* | https://github.com/ShiLab-Bioinformatics/subread RRID:SCR_009803 | |
| Software, algorithm | R v4.2.2 | The R Project for Statistical Computing | https://cran.r-project.org | |
| Software, algorithm | RStudio v2022.07.2 | RStudio Desktop | https://www.rstudio.com | |
| Software, algorithm | R package ggplot2 v3.4.3 | *Wickham, 2016* | https://ggplot2.tidyverse.org | |
| Software, algorithm | R package readxl v1.4.3 | *Wickham and Bryan, 2023a* | https://CRAN.R-project.org/package=readxl | |
| Software, algorithm | R package ggbreak v0.1.2 | *Xu et al., 2021* | https://github.com/YuLab-SMU/ggbreak RRID:SCR_014601 | |
| Software, algorithm | R package rstatix v0.7.2 | *Kassambara, 2023* | https://CRAN.R-project.org/package=rstatix | |
| Software, algorithm | R package dplyr v1.0.10 | *Wickham et al., 2023b* | https://CRAN.R-project.org/package=dplyr | |
| Software, algorithm | R package DESeq2 v1.38.3 | *Love et al., 2014* | https://github.com/mikelove/DESeq2 RRID:SCR_015687 | |
| Software, algorithm | R package ggrepel v0.9.3 | *Slowikowski et al., 2023* | https://CRAN.R-project.org/package=ggrepel | |
| Software, algorithm | R package vegan v2.6–4 | *Oksanen et al., 2022* | https://github.com/vegandevs/vegan RRID:SCR_011950 | |
| Software, algorithm | R package pairwiseAdonis v0.4 | *Arbizu, 2023* | https://github.com/pmartinezarbizu/pairwiseAdonis RRID:SCR_001905 | |

*Appendix 1 Continued*

| Reagent type (species) or resource | Designation | Source or reference | Identifiers | Additional information |
|---|---|---|---|---|
| Software, algorithm | R package gprofiler2 v0.2.2 | *Kolberg and Raudvere, 2023* | https://cran.r-project.org/web/packages/gprofiler2/index.html | |
| Software, algorithm | R package gplots v3.1.3 | *Warnes et al., 2022* | https://CRAN.R-project.org/package=gplots | |
| Software, algorithm | R package viridis v0.6.3 | *Ross et al., 2021* | https://sjmgarnier.github.io/viridis/ RRID:SCR_016696 | |
| Other | Fixable Viability Dye eFluor506 | Thermo Fisher Scientific | Cat# 65-0866-18 | (1:600) |
| Other | Nalgene Rapid-Flow PES Filter Units | Thermo Scientific | Cat# 168-0045, 165-0045, 124-0045PK | |
| Other | Vybrant DiI Cell-Labeling Solution | Invitrogen | Cat# V22885 | (1:100) |
| Other | DAPI | Sigma-Aldrich | Cat# D9542-5MG | (1:5000) |
| Other | Hoechst 33342 | Thermo Fisher Scientific | Cat# H3570 | (1:1000) |
| Other | LysoTracker | Invitrogen | Cat# L12492 | (1:1000) |

