## [Editor Report · eLife assessment]

This **fundamental** study advances our understanding of the role of bacterial-derived extracellular ATP in the pathogenesis of sepsis. The evidence supporting the conclusions is **compelling**, although not all concerns from a previous round of reviews were adequately addressed. The work will be of broad interest to researchers on microbiology and infectious diseases.

---

## [Referee Report · Reviewer #1 (Public review)]

Summary:

Extracellular ATP represents a danger-associated molecular pattern associated to tissue damage and can act also in an autocrine fashion in macrophages to promote proinflammatory responses, as observed in a previous paper by the authors in abdominal sepsis. The present study addresses an important aspect possibly conditioning the outcome of sepsis that is the release of ATP by bacteria. The authors show that sepsis-associated bacteria do in fact release ATP in a growth dependent and strain-specific manner. However, whether this bacterial derived ATP play a role in the pathogenesis of abdominal sepsis has not been determined. To address this question, a number of mutant strains of *E. coli* has been used first to correlate bacterial ATP release with growth and then, with outer membrane integrity and bacterial death. By using E. coli transformants expressing the ATP-degrading enzyme apyrase in the periplasmic space, the paper nicely shows that abdominal sepsis by these transformants results in significantly improved survival. This effect was associated to the reduction of small peritoneal macrophages and CX3CR1+ monocytes, and increase in neutrophils. To extrapolate the function of bacterial ATP from the systemic response to microorganisms, the authors exploited bacterial OMVs either loaded or not with ATP to investigate the systemic effects devoid of living microorganisms. This approach showed that ATP-loaded OMVs induced degranulation of neutrophils after lysosomal uptake, suggesting this mechanism could contribute to sepsis severity.

Strengths:

The most compelling part of the study is the analysis of *E. coli* mutants to address different aspects of bacterial release of ATP that could be pathogenically relevant during systemic dissemination of bacteria in the host.

Weaknesses:

As pointed out in the limitations of the study whether ATP-loaded OMVs could provide a mechanistic proof of the pathogenetic role of bacteria-derived ATP independently of live microorganisms in sepsis is interesting but not definitively convincing. It could be useful to see whether degranulation of neutrophils is differently induced also by apyrase-expressing vs control *E. coli* transformants. Also, the increase of neutrophils in bacterial ATP-depleted abdominal sepsis, which has better outcome than "ATP-proficient" sepsis, seems difficult to correlate to the hypothesized tissue damage induced by ATP delivered via non-infectious OMVs. Is neutrophils count affected by ATP delivered via OMVs? Probably a comparison of cytokine profiles in the abdominal fluids of *E. coli* and OMV treated animals could be helpful in defining the different responses induced by OMV-delivered vs bacterial-released ATP.

The analyses performed on OMV treated versus *E. coli* infected mice are not immediately related and difficult to combine when trying to draw a pathogenetic hypothesis for bacterial ATP in sepsis.

It's not clear why lung neutrophils were used for RNAseq.

---

## [Referee Report · Reviewer #2 (Public review)]

Summary:

In their manuscript, Daniel Spari et al. explored the dual role of ATP in exacerbating sepsis, revealing that ATP from both host and bacteria significantly impacts immune responses and disease progression.

Strengths:

The study meticulously examines the complex relationship between ATP release and bacterial growth, membrane integrity, and how bacterial ATP potentially dampens inflammatory responses, thereby impairing survival in sepsis models. Additionally, this compelling paper implies a concept that bacterial OMVs act as vehicles for the systemic distribution of ATP, influencing neutrophil activity and exacerbating sepsis severity.

Weaknesses:

(1) The researchers extracted and cultivated abdominal fluid on LB agar plates, then randomly picked 25 colonies for analysis. However, they didn't conduct 16S sequencing on the fluid itself. It's worth noting that the bacterial species present may vary depending on the individual patients. It would be beneficial if the authors could specify whether they've verified the existence of unculturable species capable of secreting high levels of Extracellular ATP.

(2) Do mice lacking commensal bacteria show a lack of Extracellular ATP following cecal ligation puncture?

(3) The authors isolated various bacteria from abdominal fluid, encompassing both Gram-negative and Gram-positive types. Nevertheless, their emphasis appeared to be primarily on the Gram-negative *E. coli*. It would be beneficial to ascertain whether the mechanisms of Extracellular ATP release differ between Gram-positive and Gram-negative bacteria. This is particularly relevant given that the Gram-positive bacterium E. faecalis, also isolated from the abdominal fluid, is recognized for its propensity to release substantial amounts of Extracellular ATP.

(4) The authors observed changes in the levels of LPM, SPM, and neutrophils in vivo. However, it remains uncertain whether the proliferation or migration of these cells is modulated or inhibited by ATP receptors like P2Y receptors. This aspect requires further investigation to establish a convincing connection.

(5) Additionally, is it possible that the observed in vivo changes could be triggered by bacterial components other than Extracellular ATP? In this research field, a comprehensive collection of inhibitors is available, so it is desirable to utilize them to demonstrate clearer results.

(6) Have the authors considered the role of host-derived Extracellular ATP in the context of inflammation?

(7) The authors mention that Extracellular ATP is rapidly hydrolyzed by ectonucleotases in vivo. Are the changes of immune cells within the peritoneal cavity caused by Extracellular ATP released from bacterial death or by OMVs?

(8) In the manuscript, the sample size (n) for the data consistently remains at 2. I would suggest expanding the sample size to enhance the robustness and rigor of the results.

---

## [Author Response]

The following is the authors’ response to the original reviews.

The points raised let us critically rethink our approach, our results, and our conclusions. Furthermore, it gave us the chance to elaborate on some critical aspects that were mentioned. With the help of the reviewers, we made some clarifications in the point-by-point responses and implemented them in the manuscript. Furthermore, we modified the figures as suggested:

- The colors in Figure 1C, D, G and H have been adapted as suggested

- We added a Figure2-figure supplement 1, which strengthens our conclusion in Figure 2

- As asked by reviewer #1 (weaknesses #3), we added the data about neutrophil numbers in the different organs (Figure 6-figure supplement 3C).

**Reviewer #1 (Public Review):**
Summary:- Extracellular ATP represents a danger-associated molecular pattern associated to tissue damage and can act also in an autocrine fashion in macrophages to promote proinflammatory responses, as observed in a previous paper by the authors in abdominal sepsis. The present study addresses an important aspect possibly conditioning the outcome of sepsis that is the release of ATP by bacteria. The authors show that sepsis-associated bacteria do in fact release ATP in a growth dependent and strain-specific manner. However, whether this bacterial derived ATP play a role in the pathogenesis of abdominal sepsis has not been determined. To address this question, a number of mutant strains of *E. coli* has been used first to correlate bacterial ATP release with growth and then, with outer membrane integrity and bacterial death. By using *E. coli* transformants expressing the ATP-degrading enzyme apyrase in the periplasmic space, the paper nicely shows that abdominal sepsis by these transformants results in significantly improved survival. This effect was associated with a reduction of peritoneal macrophages and CX3CR1+ monocytes, and an increase in neutrophils. To extrapolate the function of bacterial ATP from the systemic response to microorganisms, the authors exploited bacterial OMVs either loaded or not with ATP to investigate the systemic effects devoid of living microorganisms. This approach showed that ATP-loaded OMVs induced degranulation of neutrophils after lysosomal uptake, suggesting that this mechanism could contribute to sepsis severity.Strengths:- A strong part of the study is the analysis of *E. coli* mutants to address different aspects of bacterial release of ATP that could be relevant during systemic dissemination of bacteria in the host.

We want to thank the reviewer for recognizing this important aspect of our experimental approach.

Weaknesses:- As pointed out in the limitations of the study whether ATP-loaded OMVs provide a mechanistic proof of the pathogenetic role of bacteria-derived ATP independently of live microorganisms in sepsis is interesting but not definitively convincing. It could be useful to see whether degranulation of neutrophils is differentially induced by apyrase-expressing vs control *E. coli* transformants.

We thank the reviewer for raising several important points. In our study, we assessed local and systemic effects of released bacterial ATP. The consequences of local bacterial ATP release were assessed using an apyrase-expressing *E. coli* transformant. Locally, bacterial ATP resulted in a decrease in neutrophil numbers and we hypothesize that directly released bacterial ATP either leads to neutrophil death (e.g. via P2X7 receptor (Proietti et al., 2019)) or interferes with the recruitment of neutrophils (e.g. via P2Y receptors (Junger, 2011)).

The systemic consequences were assessed using ATP-loaded and empty OMV. We have shown that degranulation is induced by OMV-derived bacterial ATP. ATP-containing OMV are engulfed by neutrophils, reach its endolysosomal compartment and might activate purinergic receptors, which then lead to aberrant degranulation. This concept, that needs to be explored in future studies, is fundamentally different from classical purinergic signaling via directly released bacterial ATP into the extracellular space.

It is possible that neutrophil degranulation is also modulated by directly released bacterial ATP. We agree that this should be assessed in future studies. Also, the role of OMV-derived bacterial ATP should be assessed locally as well as the importance of directly released vs. OMV-mediated bacterial ATP dissected locally. Based on our measurements (Figure 4-figure supplement 1A and Figure 5C), we estimate that the effect of OMV-derived bacterial ATP might be much smaller than the effects of directly released bacterial ATP. Thus, direct ATP release might predominate locally. However, we fully agree that this has to be investigated in a future study to reconcile the different aspects of bacterial ATP signaling. A paragraph will be added to the manuscript, in which we discuss this particular issue.

- Also, the increase of neutrophils in bacterial ATP-depleted abdominal sepsis, which has better outcomes than "ATP-proficient" sepsis, seems difficult to correlate to the hypothesized tissue damage induced by ATP delivered via non-infectious OMVs.

We fully acknowledge the mentioned discrepancy. What we propose is that bacterial ATP exhibits different functions that are dependent on the release mechanism (see above). Locally, in the peritoneal cavity, neutrophil numbers are decreased by directly released bacterial ATP. Remotely, ATP is delivered via OMV and impacts on neutrophil function. We agree that, in particular, in the peritoneal cavity, both effects may play a role. However, the impact of directly released bacterial ATP seems to be dominant (see above).

We propose that neutrophils are decreased locally because of directly released bacterial ATP, which prevents efficient infection control and, therefore, impairs sepsis survival. In addition, these fewer neutrophils might even be dysregulated by the engulfment of bacterial ATP delivered via OMV, which leads to an upregulated and possibly aberrant degranulation process worsening local and remote tissue damage. We agree that in addition to neutrophil numbers, the function of local neutrophils should be assessed with and without the influence of OMV-delivered bacterial ATP. This could be done by RNA sequencing of primary neutrophils from the peritoneal cavity or neutrophil cell lines as well as degranulation assays.

- Are the neutrophils counts affected by ATP delivered via OMVs?

This is difficult to show in the peritoneal cavity where we have both, directly released bacterial ATP and OMV-derived bacterial ATP. We assessed such putative difference, however, for the systemic organs and the blood, where we did not find any differences in neutrophil numbers.

- A comparison of cytokine profiles in the abdominal fluids of E. coli and OMV treated animals could be helpful in defining the different responses induced by OMV-delivered vs bacterial-released ATP. The analyses performed on OMV treated versus *E. coli* infected mice are not closely related and difficult to combine when trying to draw a hypothesis for bacterial ATP in sepsis.

We fully agree that there are several open questions that remain to be elucidated, in particular, to differentiate the local role of directly released versus OMV-delivered bacterial ATP. In this study, we laid the foundation for future in vivo research to examine the specific role of bacterial ATP in sepsis. Such future research avenues might be to investigate the local effects of OMV-delivered bacterial ATP, and how neutrophil migration, apoptosis and degranulation are altered. We agree that exploration of the local secretory immune response and cytokine profiles are relevant to understand the different mechanisms of how bacterial ATP alters sepsis. However, such experiments should be ideally performed in systems where the source and the delivery of ATP can be modulated locally.

- Also it was not clear why lung neutrophils were used for the RNAseq data generation and analysis.

Thank you for this remark. We have chosen primary lung neutrophils for four reasons:

(1) Isolation of primary lung neutrophils allowed us to assess an in vivo response that would not have been possible with cell lines.

(2) The lung and the respiratory system are among the clinically most important organs affected during sepsis resulting in a significant cause of mortality.

(3) We show in Figure 6C that specifically in the lung, OMV are engulfed by neutrophils, which shows the relevance of the lung also in our study context.

(4) And finally, lung neutrophils were chosen to examine specifically distant and not local effects.

**Reviewer #2 (Public Review):**
Summary:- In their manuscript "Released Bacterial ATP Shapes Local and Systemic Inflammation during Abdominal Sepsis", Daniel Spari et al. explored the dual role of ATP in exacerbating sepsis, revealing that ATP from both host and bacteria significantly impacts immune responses and disease progression.Strengths:- The study meticulously examines the complex relationship between ATP release and bacterial growth, membrane integrity, and how bacterial ATP potentially dampens inflammatory responses, thereby impairing survival in sepsis models. Additionally, this compelling paper implies a concept that bacterial OMVs act as vehicles for the systemic distribution of ATP, influencing neutrophil activity and exacerbating sepsis severity.

We thank the reviewer for mentioning these key points and supporting the relevance of our study.

Weaknesses:(1) The researchers extracted and cultivated abdominal fluid on LB agar plates, then randomly picked 25 colonies for analysis. However, they did not conduct 16S rRNA gene amplicon sequencing on the fluid itself. It is worth noting that the bacterial species present may vary depending on the individual patients. It would be beneficial if the authors could specify whether they've verified the existence of unculturable species capable of secreting high levels of Extracellular ATP.

Most septic complications are caused by a limited spectrum of bacteria, belonging mainly either to the Firmicutes or the Proteobacteria phyla, including *E. coli*, *K. pneumoniae*, *S. aureus* or *E. faecalis* (Diekema et al., 2019; Mureșan et al., 2018). We validated this well documented existing evidence by randomly assessing 25 colonies. For the planned experiments, it was crucial to work with culturable bacteria; otherwise, ATP measurements, the modulation of ATP generation or loading of OMV would not have been possible. Using such culturable bacteria allowed us to describe mechanisms of ATP release.

We fully agree that hard-to-culture or unculturable bacteria might contribute significantly to septic complications. This, however, would need to be explored in future studies using extensive culturing methods (Cheng et al., 2022).

(2) Do mice lacking commensal bacteria show a lack of extracellular ATP following cecal ligation puncture?

ATP is typically secreted by many cells of the host in active and passive manners in the case of any injury, including cecal ligation and puncture (Burnstock, 2016; Dosch et al., 2018; Eltzschig et al., 2012; Idzko et al., 2014). We hypothesize that bacterial ATP is a potential priming agent at early stages of sepsis, and indeed, at such early time points, a comparison of peritoneal ATP levels between germfree and colonized mice could support our hypothesis. Future studies addressing this question must, however, correct for the different immune responses between germ-free and colonized mice. This is of utmost importance, especially for the cecal ligation and puncture model, since the cecum of germ-free mice is extremely large, making such experiments hard to control.

(3) The authors isolated various bacteria from abdominal fluid, encompassing both Gram-negative and Gram-positive types. Nevertheless, their emphasis appeared to be primarily on the Gram-negative *E. coli*. It would be beneficial to ascertain whether the mechanisms of Extracellular ATP release differ between Gram-positive and Gram-negative bacteria. This is particularly relevant given that the Gram-positive bacterium *E. faecalis*, also isolated from the abdominal fluid, is recognized for its propensity to release substantial amounts of Extracellular ATP.

We fully agree with this comment. In this paper, we used *E. coli* as our model organism to determine the principles of sepsis-associated bacterial ATP release and therefore focused on gram-negative bacteria. In addition to the direct, growth-dependent release, we found a relevant impact of OMV-delivered bacterial ATP. For this latter purpose, a gram-negative strain, in which OMV generation has been well described (Schwechheimer & Kuehn, 2015), was chosen. Recently, gram-positive bacteria have been shown to secrete ATP and OMV as well (Briaud & Carroll, 2020; Hironaka et al., 2013; Iwase et al., 2010). Given the fundamental differences in the structure of the cell wall of gram-positive bacteria and the mechanisms of OMV generation and release, future studies are required to assess the relevance of directly released and OMV-delivered ATP in gram-positive bacteria.

(4) The authors observed changes in the levels of LPM, SPM, and neutrophils in vivo. However, it remains uncertain whether the proliferation or migration of these cells is modulated or inhibited by ATP receptors like P2Y receptors. This aspect requires further investigation to establish a convincing connection.

We fully agree with this comment. The decrease in LPM and the consequential predomination of SPM have been well described after inflammatory stimuli in the context of the macrophage disappearance reaction (Ghosn et al., 2010). Also, it has been shown that purinergic signaling modulates infiltration of neutrophils and can lead to cell death as a consequence of P2Y and P2X receptor activation (Junger, 2011; Proietti et al., 2019). In our study, we propose that intracellular purinergic receptors contribute to neutrophil function during sepsis. After introducing the general principles and fundaments of bacterial ATP with our studies, we fully agree that additional experiments need to address downstream purinergic receptor activation. That, however, would go beyond the scope of our study.

(5) Additionally, is it possible that the observed in vivo changes could be triggered by bacterial components other than Extracellular ATP? In this research field, a comprehensive collection of inhibitors is available, so it is desirable to utilize them to demonstrate clearer results.

This question is of utmost importance and defined the choice of our model and experimental approach. When we started the project, we used two different *E. coli* mutants that release low (ompC) and high (eaeH) amounts of ATP. However, the limitation of this approach is that these are different bacteria, which may also differ in the components they secrete or the surface proteins they express. We, therefore, decided against that approach. With the approach we finally used (same bacterium, just with and without ATP), we aimed to minimize the influence of non-ATP bacterial components.

(6) Have the authors considered the role of host-derived Extracellular ATP in the context of inflammation?

Yes, the role of host-derived extracellular ATP in inflammation and sepsis is well-established with contradictory results (Csóka et al., 2015; Ledderose et al., 2016). This conflicting data was the rationale to test the relevance of bacterial ATP. We suggest that bacterial ATP is essential in the early phase of sepsis when bacteria invade the sterile compartment and before efficient host response, including the eukaryotic release of ATP, is established.

(7) The authors mention that Extracellular ATP is rapidly hydrolyzed by ectonucleotases in vivo. Are the changes of immune cells within the peritoneal cavity caused by Extracellular ATP released from bacterial death or by OMVs?

This is a relevant question that was also asked by reviewer #1, and we answered it in detail above (weaknesses comment #1 and #2). From our ATP measurements (Figure 4-figure supplement 1A and Figure 5C), we conclude that locally, the role of directly released bacterial ATP (extracellular) predominates over OMV-derived bacterial ATP. Furthermore, the mechanisms between directly released and OMV-derived bacterial ATP (within OMV, engulfed and transported to the endolysosomal compartment) are different, and especially extracellular ATP has been described to lead to apoptosis via P2X7 signaling.

(8) In the manuscript, the sample size (n) for the data consistently remains at 2. I would suggest expanding the sample size to enhance the robustness and rigor of the results.

Two biological replicates (independent cultures) were only used for the bacteria cultures in Figure 1, Figure 2, and Figure 3, which achieved similar results and the standard deviation remained very small, indicating its robustness. In the in vitro experiments in Figure 5 we used a sample size of 6 (three biological replicates measured in technical duplicates), since we saw bigger deviations in our measurements. For the in vivo experiments, we always used 5 or more animals in at least two independent experiments.

**Reviewer #2 (Recommendations For The Authors):**
(9). Line 37: 11 million sepsis-related deaths were reported "in" 2017.The passage has been corrected as suggested.(10) By the way, the similar colors used in Figure 1C and G are too chaotic, making it difficult to distinguish.

We agree, the colors have been adapted.

**Author response image 2. sa3fig2:** 

(11). All "in vivo" and "in vitro" should be italicized.

We italicized all of them.

(12). The title of Figure 4 is confusing: "Impairs sepsis outcome in vivo?" Could you make it more specific?

We agree, the title has been rephrased:

“Bacterial ATP reduces neutrophil counts and reduces survival in a mouse model of abdominal sepsis.”

(13) Line 314-316: The sentence "Potentially, despite the lack of a transporter, ATP may similarly to eukaryotic cells leak (Yegutkin et al., 2006) across the inner membrane into the periplasmic space that lacks the enzymes for ATP generation." sounds odd.

This passage was reformulated in the manuscript.

“Despite the lack of a transporter, ATP may leak across the inner membrane into the periplasmic space. Such leakage may be similar to baseline leakage in eukaryotic cells (Yegutkin et al., 2006).”

(14) The numerical notation in the paper is odd: sometimes it uses a prime symbol as a superscript (such as line 504), and sometimes it does not (such as line 421). Should it be standardized to "3,200" and "150,000"?

Thank you for this remark. The numbers have been standardized throughout the manuscript.

(15) Line "0.4 mm EP cuvettes" should be "0.4 cm EP cuvettes"

The specified passage has been corrected as suggested.

References

Briaud, P., & Carroll, R. K. (2020). Extracellular Vesicle Biogenesis and Functions in Gram-Positive Bacteria. Infection and Immunity, 88(12), 10.1128/iai.00433-20. https://doi.org/10.1128/iai.00433-20

Burnstock, G. (2016). P2X ion channel receptors and inflammation. Purinergic Signalling, 12(1), 59–67. https://doi.org/10.1007/s11302-015-9493-0

Cheng, A. G., Ho, P.-Y., Aranda-Díaz, A., Jain, S., Yu, F. B., Meng, X., Wang, M., Iakiviak, M., Nagashima, K., Zhao, A., Murugkar, P., Patil, A., Atabakhsh, K., Weakley, A., Yan, J., Brumbaugh, A. R., Higginbottom, S., Dimas, A., Shiver, A. L., … Fischbach, M. A. (2022). Design, construction, and in vivo augmentation of a complex gut microbiome. Cell, 185(19), 3617-3636.e19. https://doi.org/10.1016/j.cell.2022.08.003

Csóka, B., Németh, Z. H., Törő, G., Idzko, M., Zech, A., Koscsó, B., Spolarics, Z., Antonioli, L., Cseri, K., Erdélyi, K., Pacher, P., & Haskó, G. (2015). Extracellular ATP protects against sepsis through macrophage P2X7 purinergic receptors by enhancing intracellular bacterial killing. The FASEB Journal, 29(9), 3626–3637. https://doi.org/10.1096/fj.15-272450

Diekema, D. J., Hsueh, P.-R., Mendes, R. E., Pfaller, M. A., Rolston, K. V., Sader, H. S., & Jones, R. N. (2019). The Microbiology of Bloodstream Infection: 20-Year Trends from the SENTRY Antimicrobial Surveillance Program. Antimicrobial Agents and Chemotherapy, 63(7), e00355-19. https://doi.org/10.1128/AAC.00355-19

Dosch, M., Gerber, J., Jebbawi, F., & Beldi, G. (2018). Mechanisms of ATP Release by Inflammatory Cells. International Journal of Molecular Sciences, 19(4), 1222. https://doi.org/10.3390/ijms19041222

Eltzschig, H. K., Sitkovsky, M. V., & Robson, S. C. (2012). Purinergic Signaling during Inflammation. New England Journal of Medicine, 367(24), 2322–2333. https://doi.org/10.1056/NEJMra1205750

Ghosn, E. E. B., Cassado, A. A., Govoni, G. R., Fukuhara, T., Yang, Y., Monack, D. M., Bortoluci, K. R., Almeida, S. R., Herzenberg, L. A., & Herzenberg, L. A. (2010). Two physically, functionally, and developmentally distinct peritoneal macrophage subsets. Proceedings of the National Academy of Sciences, 107(6), 2568–2573. https://doi.org/10.1073/pnas.0915000107

Hironaka, I., Iwase, T., Sugimoto, S., Okuda, K., Tajima, A., Yanaga, K., & Mizunoe, Y. (2013). Glucose Triggers ATP Secretion from Bacteria in a Growth-Phase-Dependent Manner. Applied and Environmental Microbiology, 79(7), 2328–2335. https://doi.org/10.1128/AEM.03871-12

Idzko, M., Ferrari, D., & Eltzschig, H. K. (2014). Nucleotide signalling during inflammation. Nature, 509(7500), 310–317. https://doi.org/10.1038/nature13085

Iwase, T., Shinji, H., Tajima, A., Sato, F., Tamura, T., Iwamoto, T., Yoneda, M., & Mizunoe, Y. (2010). Isolation and Identification of ATP-Secreting Bacteria from Mice and Humans. Journal of Clinical Microbiology, 48(5), 1949–1951. https://doi.org/10.1128/JCM.01941-09

Junger, W. G. (2011). Immune cell regulation by autocrine purinergic signalling. Nature Reviews Immunology, 11(3), 201–212. https://doi.org/10.1038/nri2938

Ledderose, C., Bao, Y., Kondo, Y., Fakhari, M., Slubowski, C., Zhang, J., & Junger, W. G. (2016). Purinergic Signaling and the Immune Response in Sepsis: A Review. Clinical Therapeutics, 38(5), 1054–1065. https://doi.org/10.1016/j.clinthera.2016.04.002

Mureșan, M. G., Balmoș, I. A., Badea, I., & Santini, A. (2018). Abdominal Sepsis: An Update. The Journal of Critical Care Medicine, 4(4), 120–125. https://doi.org/10.2478/jccm-2018-0023

Proietti, M., Perruzza, L., Scribano, D., Pellegrini, G., D’Antuono, R., Strati, F., Raffaelli, M., Gonzalez, S. F., Thelen, M., Hardt, W.-D., Slack, E., Nicoletti, M., & Grassi, F. (2019). ATP released by intestinal bacteria limits the generation of protective IgA against enteropathogens. Nature Communications, 10(1), Article 1. https://doi.org/10.1038/s41467-018-08156-z

Schwechheimer, C., & Kuehn, M. J. (2015). Outer-membrane vesicles from Gram-negative bacteria: Biogenesis and functions. Nature Reviews Microbiology, 13(10), 605–619. https://doi.org/10.1038/nrmicro3525